# Versatile synthesis of metal-compound based mesoporous Janus nanoparticles

Yan Yu[1], Runfeng Lin[1], Hongyue Yu[1], Minchao Liu[1], Enyun Xing[1], Wenxing Wang[1], Fan Zhang [1], Dongyuan Zhao [1] & Xiaomin Li [1] ✉

The construction of mesoporous Janus nanoparticles (mJNPs) with controllable components is of great significance for the development of sophisticated nanomaterials with synergistically enhanced functionalities and applications. However, the compositions of reported mJNPs are mainly the functionally inert $SiO_2$ and polymers. The universal synthesis of metal-compound based mJNPs with abundant functionalities is urgently desired, but remains a substantial challenge. Herein, we present a hydrophilicity mediated interfacial selective assembly strategy for the versatile synthesis of metal-compound based mJNPs. Starting from the developed silica-based mJNPs with anisotropic dual-surface of hydrophilic $SiO_2$ and hydrophobic organosilica, metal precursor can selectively deposit onto the hydrophilic $SiO_2$ subunit to form the metal-compound based mJNPs. This method shows good universality and can be used for the synthesis of more than 20 kinds of metal-compound based mJNPs, including alkali-earth metal compounds, transition metal compounds, rare-earth metal compounds etc. Besides, the composition of the metal-compound subunit can be well tuned from single to multiple metal elements, even high-entropy complexes. We believe that the synthesis method and obtained new members of mJNPs provide a very broad platform for the construction and application of mJNPs with rational designed functions and structures.

Janus nanoparticles (JNPs) are colloidal patchy nanocomposites consisting of at least two distinctive subunits, which exhibit multiple different surface chemistries, functionalities, and anisotropy-derived new properties[1–6]. Based on this unique asymmetric structure, the JNPs can offer unlimited possibilities in either promoting properties of their individual subunits or integrating various functional components within one single structures, even generating new properties and functions[7–9]. The fascinating properties and functions of the JNPs depend not only on their structures, but also on their compositions, the spatial distributions of each subunit and the interfaces between the subunits[10–12]. Therefore, the rational design and synthesis of Janus nanostructures with controllable compositions and functionalities is of great importance not only for the realization of improved performance in specific applications, but also for improving our understanding of fundamental structure-property-function relationships.

To date, numerous sophisticated JNPs have been synthesized based on the building blocks of nanocrystals[13–16], metal-organic frameworks nanoparticles[17–20], mesoporous nanoparticles[21–23], polymer nanoparticles[24–26] and so on. Among them, the mesoporous nanoparticles with high surface areas, tunable pore sizes and structures, controllable framework compositions are becoming an exciting building block for the construction of mesoporous Janus nanoparticles (mJNPs)[27,28]. After less than a decade of rapid development, a variety of mJNPs has been explored, including multi-compartment mesoporous silica ($mSiO_2$)[29], dumbbell-like RF&PMO nanoparticles (RF means resorcinol formaldehyde, PMO means periodic mesoporous

[1]Department of Chemistry, Shanghai Stomatological Hospital & School of Stomatology, State Key Laboratory of Molecular Engineering of Polymers, iChem, Shanghai Key Laboratory of Molecular Catalysis and Innovative Materials, Fudan University, Shanghai 200433, China. ✉e-mail: lixm@fudan.edu.cn

organosilica)[30], multipods RF&PMO[31], multipodal hybrid PMO[32], Janus mSiO$_2$&PMO[33], and rSiO$_2$&PMO (rSiO$_2$ refers to SiO$_2$ nanoparticle with rough surface)[34]. The mJNPs can not only provide anisotropic surfaces for site-specific functionalization, independent storage spaces for guest molecules, but also possess multiple surfaces and unique heterojunctions for the enhanced matter/energy exchange efficiency with external environments[27,35]. These features make the mJNPs have great prospects in multi-drugs delivery[36,37], active cargo delivery[38,39], nanomedicine[40], biphasic cascade catalysis[41,42] and so on. However, the composition of the obtained mJNPs is mainly limited to functionally inert SiO$_2$ and polymer[6,35]. The reported synthesis methods are difficult to apply to the construction of metal-compound based mJNPs, because of the following reasons. First, due to the lack of driving force for anisotropic assembly of metal-compound, the metal precursors easily aggregate into phase-separated nanoparticles before they can anisotropically assemble into Janus nanostructure. Second, it is still a great challenge to generically synthesize the metal-compound based mJNPs, because the hydrolysis rates of various metal precursors are quite different. Considering their abundant functionalities, the construction of metal-compound based mJNPs with controllable architecture and composition is urgently desired for the synergistically enhanced functions and applications, but has rarely been demonstrated.

In this work, a library of metal-compound based mJNPs (M-mJNPs) with controllable architecture, composition and function was constructed via a hydrophilicity-mediated interfacial selective growth strategy. By introducing asymmetric SiO$_2$&PMO nanostructure with dual-compartments of hydrophilic SiO$_2$ and hydrophobic PMO as a template, we demonstrate the selective assembly of metal compound on SiO$_2$ subunit, forming the new M-mJNPs derivatives. The architecture of the obtained M-mJNPs can be regulated by tuning the spatial position and shape of the hydrophilic domain in a pristine template (Fig. 1a, i). In addition, the composition of the metal-compound subunit of the M-mJNPs can be rationally tuned from single to multiple metal elements, even high-entropy complexes (including but not limited to the alkaline-earth metal of Ca and Mg, transition metal of Mn, Fe, Co, Ni, Cu, Zn, Cd, and almost all of the rare-earth elements). The expansion of composition can bring rich functionalities to M-mJNPs for a wide range of applications. As a proof of concept, the Fe-mJNPs-GOx (GOx is glucose oxidase grafted on the opposite side of Fe based subunit) is used as a spatially asymmetric cascade nanocatalyst for enhanced chemodynamic therapy (CDT), in which the GOx grafted subunit can effectively deplete glucose in tumor cells, and meanwhile produce a considerable amount of H$_2$O$_2$ for subsequent Fenton reaction under the catalysis of Fe based subunit in the tumor microenvironment. Taking advantage of the spatial isolation of GOx grafted subunit and Fe based subunit, the cascade catalytic efficiency of the mJNPs nanocatalysts is greatly increased, thus realizing remarkably efficient CDT for cancer cell killing and tumor restrain.

## Results and discussion

### Synthesis of Ni-mJNPs via interfacial selective assembly strategy

The uniform metal-compound based mJNPs were synthesized via hydrophilicity mediated interfacial selective growth strategy. Janus sSiO$_2$&rPMO template consisting of a hydrophilic SiO$_2$ nanosphere (sSiO$_2$) and a hydrophobic rod-shaped PMO (rPMO) was synthesized via anisotropic growth strategy[33]. Transmission electron microscope (TEM) image of the obtained sSiO$_2$&rPMO template with anisotropic dual-surface of hydrophilic and hydrophobic (Supplementary Fig. 1) shows good dispersity and distinctive asymmetric structure consisting of a spherical sSiO$_2$ head (~120 nm) and a rod-shaped tail (~120 nm in length and ~110 nm in diameter).

As shown in Fig. 1a, by using sSiO$_2$&rPMO as templates, the metal-compound are expected to selectively assemble on the hydrophilic sSiO$_2$ domain after the introduction of the metal precursors. We use Ni(NO$_3$)$_2$ as a typical metal precursor to perform the hydrophilicity-mediated interfacial selective assembly, and the Ni-sSiO$_2$&rPMO mJNPs were successfully synthesized. As shown in the scanning electron microscope (SEM) image (Supplementary Fig. 2), the obtained Ni-sSiO$_2$&rPMO mJNPs show distinctive asymmetric morphology consisting of a spherical head with rough surface and a rod-like tail with smooth surface. TEM images (Fig. 1b, c) of the obtained Ni-sSiO$_2$&rPMO mJNPs clearly reveals that the sSiO$_2$ domain of the pristine template is wrapped by a layer of nanosheets to form a core-shell structured head, while the rPMO tail remains uncovered. The high-angle annular dark field imaging in the scanning TEM (HAADF-STEM) and energy dispersive X-ray Spectroscopy (EDS) mapping images show that Si and O elements are uniformly distributed all over the mJNPs, while Ni element is mainly distributed on the spherical head (Fig. 1d). Besides, the porosity of the Ni-sSiO$_2$&rPMO mJNPs can be clearly observed in the TEM and SEM images. The pore size distribution analysis via nitrogen sorption measurement shows that obtained mJNPs possess dual mesopores at about 3 and 9 nm (Fig. 1e), which can be attributed to the mesopores in the pristine sSiO$_2$&rPMO template (Supplementary Fig. 3) and piled mesopores in the Ni-based subunit of mJNPs. The well retained 3 nm sized mesopore inherited from the pristine template indicates that there is no metal compounds deposition inside the mesoporous structure in the Ni-sSiO$_2$&rPMO mJNPs. The X-ray diffraction (XRD) pattern of the obtained Ni-sSiO$_2$&rPMO mJNPs (Fig. 1h) show three distinct peaks at 22.7°, 33.4° and 59.9°, which is corresponding to the (006), (101) and (110) crystal planes of rhombohedral Ni(OH)$_2 \cdot$0.5H$_2$O (JCPDS no. 00-038-0715). The Ni 2p X-ray photoelectron spectroscopy (XPS) spectrum of the obtained Ni-sSiO$_2$&rPMO mJNPs further confirms the composition of Ni(OH)$_2$ (Supplementary Fig. 4a).

The crystalline Ni(OH)$_2$ can be easily transformed into NiO after calcination at 600 °C in ambient atmosphere. Observing from the TEM image (Fig. 1f) and elemental mapping of the calcined Ni-sSiO$_2$&rPMO mJNPs, the Janus nanostructure is retained very well after calcination and the Ni element is still only selectively distributed on the surface of the head compartment of the mJNPs (Supplementary Fig. 5). The corresponding XRD pattern (Fig. 1h), XPS spectrum (Supplementary Fig. 4b) and high-resolution TEM (HRTEM) image (Fig. 1g) reveal the highly crystalline cubic structure of NiO subunit. Besides, the nitrogen sorption isotherms (Supplementary Fig. 6) also confirm the well retained mesoporous structure of the calcined Ni-sSiO$_2$&rPMO mJNPs.

The architecture of the Ni-sSiO$_2$&rPMO mJNPs can be regulated by tuning the spatial location and shape of hydrophilic SiO$_2$ domain in the pristine template. Other than the selective growth on the spherical head, the metal compounds can selectively assemble on the rod-shaped tail by switching the hydrophilicity of the head and tail of the pristine template (Fig. 1i). For example, utilizing an asymmetric sPMO&rSiO$_2$ template with a hydrophobic spherical PMO (sPMO) head and a hydrophilic rod-shaped SiO$_2$ (rSiO$_2$) tail (Supplementary Fig. 7), the Ni(OH)$_2$ nanosheets can selectively assemble on the rod-shaped tail of the template, resulting in the formation of sPMO&rSiO$_2$-Ni mJNPs. As shown in the SEM and TEM images (Fig. 1j, k, Supplementary Fig. 8), the obtained sPMO&rSiO$_2$-Ni mJNPs also show a distinct asymmetric structure consisting of a smooth spherical head and a rough rod-like tail wrapped by a layer of nanosheets. EDS mapping images clearly depict the Janus structure of sPMO&rSiO$_2$-Ni, in which the Ni element is selectively distributed around the rod-like tail (Fig. 1l).

### General applicability of the interfacial selective assembly strategy

By simply replacing the metal precursors, this hydrophilicity-mediated selective assembly strategy can be easily extended to the synthesis of other transition and rare-earth metal-compound based mJNPs. A variety of representative M-sSiO$_2$&rPMO mJNPs (M=Mn, Co, Cu, Zn, Fe, Cd, Y, Gd, Ce, Yb) with controllable compositions were synthesized (Fig. 2,

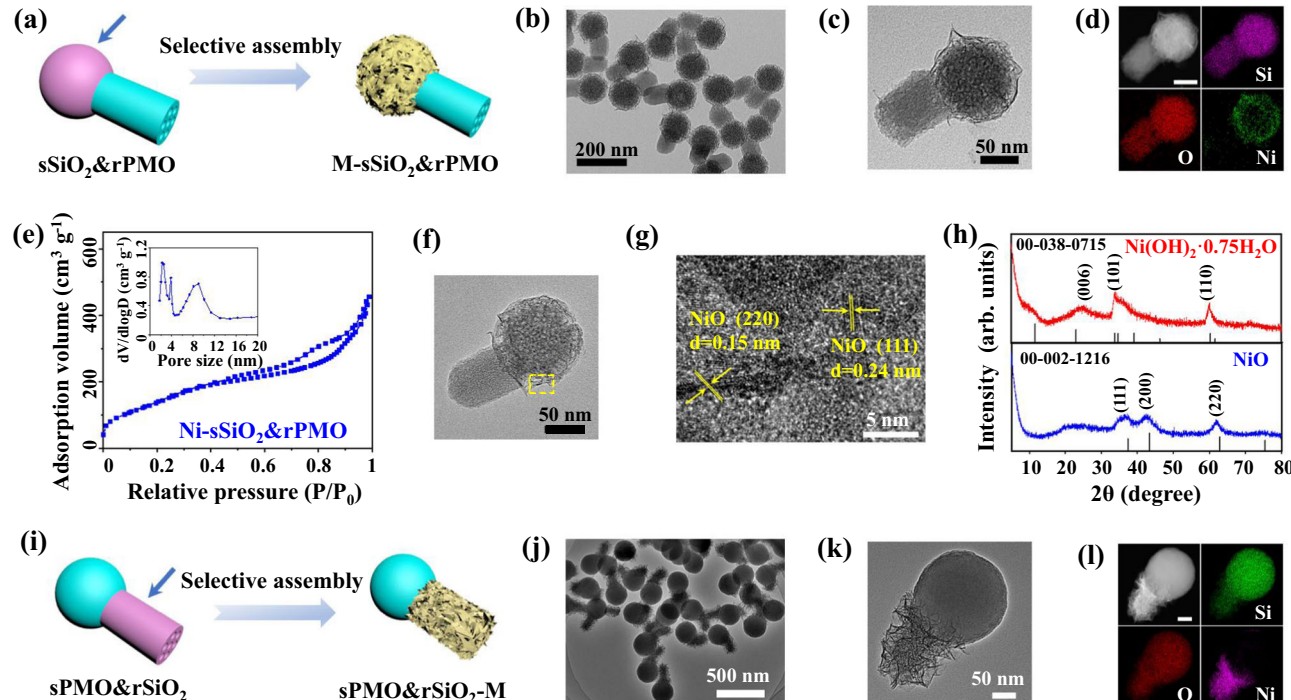

**Fig. 1 | The Ni-mJNPs with controllable architectures. a** Schematic illustration of the selective assembly of metal compound on the pristine sSiO$_2$&rPMO template with spherical SiO$_2$ (sSiO$_2$) head and rod-shaped PMO (rPMO) tail. **b, c** TEM, **d** HAADF-STEM and EDS elemental mapping images of the obtained Ni-sSiO$_2$&rPMO mJNPs with spherical Ni-rSiO$_2$ head and rPMO tail. **e** N$_2$ sorption isotherm and the corresponding pore size distribution of the obtained Ni-sSiO$_2$&rPMO mJNPs. **f** TEM and **g** HRTEM images of the calcinated Ni-sSiO$_2$&rPMO mJNPs. **h** XRD patterns of Ni-sSiO$_2$&rPMO mJNPs before and after calcination. **i** Schematic illustration of the selective assembly of metal compound on the rod-shaped SiO$_2$ (rSiO$_2$) domain of pristine sPMO&rSiO$_2$ templete (sPMO means spherical PMO). **j–l** TEM, HAADF-STEM and EDS elemental mapping images of sPMO&rSiO$_2$-Ni mJNPs with spherical sPMO head and rSiO$_2$-Ni tail. Source data are provided as a Source Data file.

Supplementary Fig. 9–11). Almost all the rare-earth metal compounds can selectively grow on the hydrophilic SiO$_2$ domain of the pristine template to form the rare-earth based mJNPs. All the obtained M-mJNPs with different compositions are monodispersed, and possess well-defined Janus heterostructure consisting of spherical core-shell structured heads and uncovered rod-like tails (Supplementary Fig. 9–11). Due to the different hydrolysis manners of different metal precursors, the metal compounds on the sSiO$_2$ head are composed by the thin nanosheets, short nanowires or small nanoparticles (Supplementary Fig. 9g–l). According to XPS spectra (Supplementary Fig. 12–15) and XRD patterns (Supplementary Fig. 9m–r), these metal compounds can be divided into three types: crystalline hydroxide (e.g. Ni(OH)$_2$, Zn(OH)$_2$), crystalline hydroxysilicate (e.g. Co$_3$(OH)$_4$Si$_2$O$_5$), and amorphous compounds (e.g. Mn-, Cu-, Y- and Gd- based compounds).

After calcination, the metal compounds can further transform into crystalline metal oxides (Fig. 2). TEM images show that all the obtained metal oxides based mJNPs maintained good dispersity and asymmetric nanostructure after calcination. The EDS mapping images further demonstrate that all the metal elements are homogeneously distributed at the head compartment of the mJNPs. And the pore size distribution analysis confirms well reserved mesoporous structure in the mJNPs (Supplementary Fig. 16). According to the XRD patterns (Fig. 2g), the crystal phases of the metal-compound subunits are determined to be Mn$_2$O$_3$ (JCPDS no. 00-002-0896), Co$_3$O$_4$ (JCPDS no. 00-001-1152), CuO (JCPDS no. 01-080-1268), ZnO (JCPDS no. 00-036-1451), Y$_2$O$_3$ (JCPDS no. 00-001-0831) and Gd$_2$O$_3$ (JCPDS no. 00-043-1014), respectively. The XPS analysis (Supplementary Figs. 12–15) and lattice fringes in HRTEM images (Supplementary Fig. 17) are well matched with the XRD patterns. In addition to transition and rare-earth metal compounds, this selective assembly strategy can also be readily

extended to the synthesis of alkaline-earth metal-compound based mJNPs. For example, the Ca-sSiO$_2$&rPMO (Fig. 2f) and Mg-sSiO$_2$&rPMO (Fig. 2g) mJNPs are also successfully synthesized, in which alkaline-earth metal-compound subunits on the sSiO$_2$ heads are composed by granular CaCO$_3$ (JCPDS no. 00-024-0030) or lamellar MgSiO$_3$ (JCPDS no. 00-047-1750), respectively. The phase of CaCO$_3$ and MgSiO$_3$ in Ca-mJNPs and Mg-mJNPs are further validated by XPS analysis (Supplementary Fig. 18). The metal contents in these representative metal-compound based mJNPs shown in Fig. 2 were measured by inductively coupled plasma-optical emission spectroscopy (ICP-OES) and listed in Supplementary Table 1.

Furthermore, this hydrophilicity mediated interfacial selective growth strategy is also applicable to the synthesis of M$_x$-sSiO$_2$&rPMO mJNPs with multiple metal elements (subscript x represents the number of metal-element types). The binary M$_2$-sSiO$_2$&rPMO mJNPs can be easily obtained by simply using two kinds of metal precursors (Supplementary Fig. 19). By further increasing the number of metal-precursor types, mJNPs with more complicated metal-compound subunits, even high-entropy complex subunits can be synthesized (Supplementary Fig. 20). For example, Mn/Co/Ni-sSiO$_2$&rPMO mJNPs with ternary metal-compound subunits can be synthesized by using the mixture of MnCl$_2$, Co(NO$_3$)$_2$ and Ni(NO$_3$)$_2$ as mixed metal precursor, in which the spherical silica is fully wrapped by metal-compound subunit (Supplementary Fig. 21). After calcination, the amorphous Mn/Co/Ni-sSiO$_2$&rPMO can be transformed into crystalline ternary metal-oxides based mJNPs, and the morphology is well retained (Fig. 3b). The XRD pattern of the calcined Mn/Co/Ni-sSiO$_2$&rPMO mJNPs is similar to that of Ni-sSiO$_2$&rPMO (Fig. 3c). The diffraction peaks slightly shift to the low-angle direction and accompanied by peak broadening, which may be attributed to the partial substitution of nickel ions in crystalline NiO by manganese and cobalt

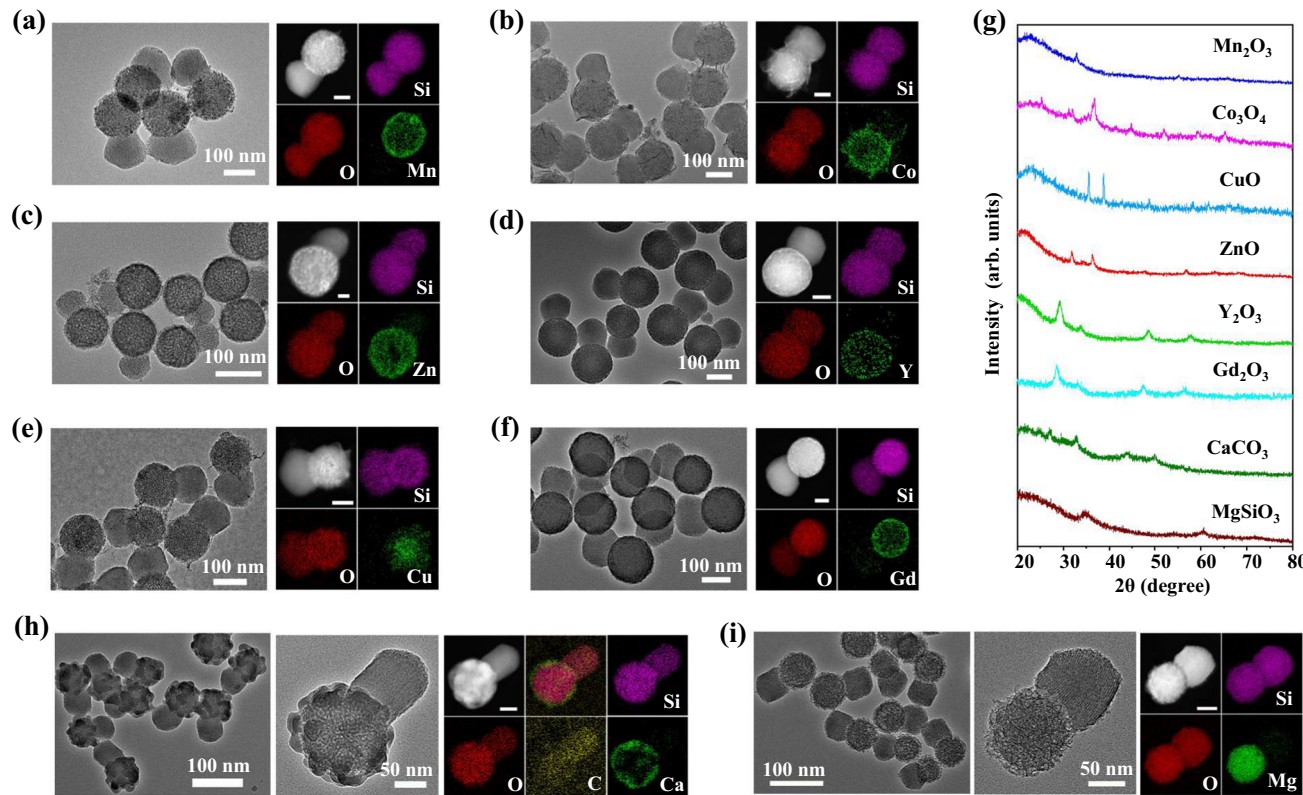

**Fig. 2 | The generality of the strategy for the synthesis of metal-compound based mJNPs. a–f**, **h**, **i** TEM, HAADF-STEM, and EDS elemental mapping of the obtained metal-compound based mJNPs. **a** Mn-sSiO$_2$&rPMO, **b** Co-sSiO$_2$&rPMO, **c** Cu-sSiO$_2$&rPMO, **d** Zn-sSiO$_2$&rPMO, **e** Y-sSiO$_2$&rPMO, **f** Gd-sSiO$_2$&rPMO, **h** Ca-sSiO$_2$&rPMO and **i** Mg-sSiO$_2$&rPMO. The scale bars in the EDS mapping images are 50 nm. **g** XRD patterns of representative metal-compound based mJNPs. Source data are provided as a Source Data file.

atoms. According to the EDS mapping images (Fig. 3d) of the Mn/Co/Ni-sSiO$_2$&rPMO mJNPs, the Mn, Co and Ni elements are homogeneously distributed in the head compartment of the mJNPs without any aggregation, indicating that there is no phase separation in the ternary metal-compounds based mJNPs.

Besides, a M$_9$-sSiO$_2$&rPMO mJNPs with 9 kinds of rare-earth metal elements (Tm, Y, La, Pr, Tb, Nd, Ho, Dy, and Lu) was synthesized (Fig. 3e–h, Supplementary Fig. 22). Similarly, after calcination, the monodispersed Janus nanostructure is also well retained. The XRD patterns show that nonuple metal-compounds based mJNPs transformed from amorphous to crystalline after calcination, and the diffraction peaks slightly shift to lower angle and have a significant peak broadening compared to Y-sSiO$_2$&PMO mJNPs (Fig. 3g). The EDS mapping images (Fig. 3h) clearly reveal that all the rare-earth elements are uniformly distributed on the spherical head of the obtained M$_9$-sSiO$_2$&rPMO mJNPs, indicating the formation of a high-entropy subunit. These results show that the composition of the metal-compound subunits of mJNPs can be rationally tuned from single to multiple metal elements, even high-entropy complexes, demonstrating the generality of this synthetic strategy.

### Hydrophilicity mediated interfacial selective growth mechanism for the synthesis of M-mJNPs

A hydrophilicity mediated interfacial selective growth mechanism is proposed for the synthesis of the above-mentioned metal-compound based mJNPs (Fig. 4a). The above results have demonstrated that the metal compounds preferentially grow on SiO$_2$ domain of the pristine SiO$_2$&PMO template. We believe that this selective growth manner is mainly due to the difference in hydrophilicity of the SiO$_2$ and PMO surfaces. To verify this assumption, we further synthesized two kinds of templates with Janus morphology, but isotropic surface properties,

i.e., sSiO$_2$&rSiO$_2$ with isotropic hydrophilic-surface and sPMO&rPMO with isotropic hydrophobic-surface (Supplementary Fig. 23). Under the same synthesis conditions, the selective assembly behavior of metal compounds does not occur, and sSiO$_2$&rSiO$_2$ nanoparticles are completely encapsulated by metal-compound shell (Fig. 4b–I). In comparison, the metal compounds cannot grow on the sPMO&rPMO template, but form the irregular phase-separated nanosheets (Fig. 4b–II). The contact angle of water droplets on SiO$_2$ surface (~6.8° for spherical SiO$_2$, ~7.6° for rod-like SiO$_2$ nanoparticles) is much smaller than that of on PMO surface (~35° for spherical PMO, ~40° for rod-like PMO nanoparticles) (Fig. 4c, Supplementary Fig. 24), confirming the different hydrophilicity between SiO$_2$ and PMO domains. So, we assume that the selective growth of metal compounds is not related to the morphology of the template, but result from the anisotropic distribution of the hydrophilicity.

First, we calculated the electrostatic potential distribution of SiO$_2$ and PMO (ethyl-bridged organosilica) and binding energy between water molecule and SiO$_2$ or ethyl-bridged organosilica surface (Supplementary Data 1, Fig. 4d, e), respectively, by using density functional theory (DFT). By partitioning the total electron charge density into the contributions from individual atoms via Mulliken analysis, it is revealed that the electrostatic charge numbers on O atoms in SiO$_2$ are lower than that of the O atoms in PMO (−0.35), especially the O atom linking the two Si atoms (−0.76). So, the binding energy of water molecule with SiO$_2$ is about −54.72 kcal mol$^{-1}$, which is higher than that of with PMO (−42.98 kcal mol$^{-1}$). In addition, from the view point of reaction kinetics, the density of hydroxyl groups on the surface of SiO$_2$ (~8.68 per nm$^2$) is much higher than that of on the PMO surface (~4.09 per nm$^2$). The high hydroxyl group density on the SiO$_2$ further promote the probability of the interaction between water molecules and SiO$_2$.

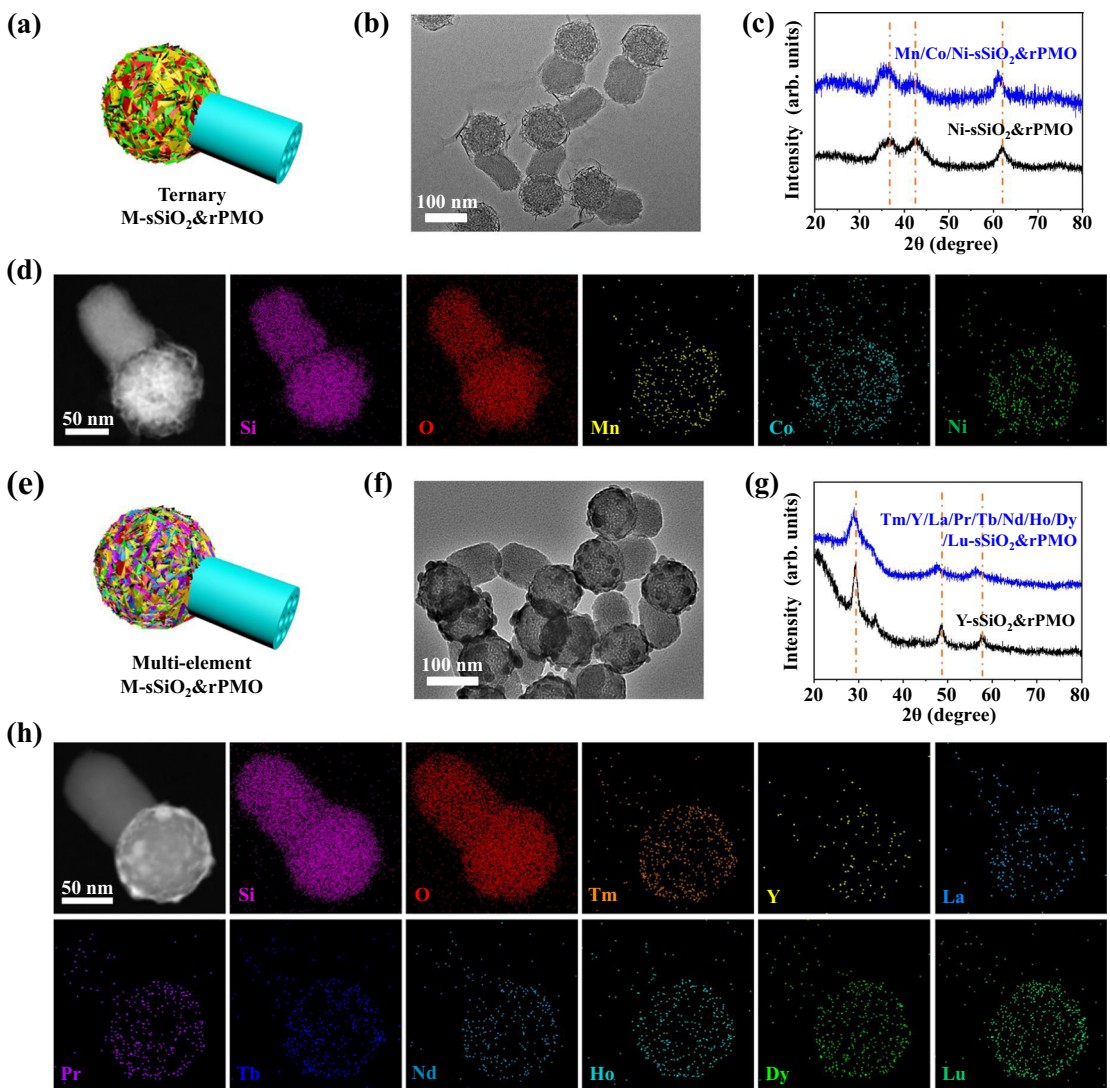

**Fig. 3 | The synthesis of metal-compound based mJNPs with multiple metal elements. a** Schematic illustration of $M_3$-sSiO$_2$&rPMO mJNPs with ternary metal elements. **b** TEM image, **c** XRD patterns and **d** HAADF-STEM and EDS elemental mapping of Mn/Co/Ni-sSiO$_2$&rPMO mJNPs with ternary metal compounds. **e** Scheme illustration of $M_x$-sSiO$_2$&rPMO mJNPs with multiple metal elements. **f** TEM image, **g** XRD patterns and **h** HAADF-STEM and EDS elemental mapping of $M_9$-sSiO$_2$&rPMO mJNPs with nonuple rare-earth elements of Tm, Y, La, Pr, Tb, Nd, Ho, Dy, and Lu. Source data are provided as a Source Data file.

The hydrolysis of metal ions in aqueous solution can be represented by the following reaction equation:

$$[M(H_2O)_m]^{Z+} + H_2O \rightleftharpoons [M(OH)(H_2O)_{m-1}]^{(Z-1)+} + H_3O^+ \qquad (1)$$

Metal ions in aqueous solution behave as Lewis acids. The positive charge on the metal ion draws electron density from the O–H bond in the water. This increases the bond's polarity, making it easier for the O–H bond to break and release protons. Herein, we chose [Ni(H$_2$O)$_4$]$^{2+}$ as the model and calculated the electrostatic potential distribution of this molecule on the surface of silica and ethyl-bridged organosilica molecules (Fig. 4f), respectively. Compared with [Ni(H$_2$O)$_4$]$^{2+}$ molecules absorbed on the organosilica surface, the exposed hydrogen atoms of the [Ni(H$_2$O)$_4$]$^{2+}$ molecules absorbed on the silica surface have higher peak electrostatic potential energy (-9.1 eV), which makes this hydrogen atom easier to be dissociated. Thus, compared with organosilica, [Ni(H$_2$O)$_4$]$^{2+}$ on the silica surface is easier to be hydrolyzed, thereby facilitating the subsequent selective deposition of metal hydroxides. If [Ni(H$_2$O)$_4$]$^{2+}$ has released a proton prior to contact with pristine template, the [Ni(OH)(H$_2$O)$_3$]$^+$ is formed and interact with silanol groups on the surface of SiO$_2$ and PMO. Similarly, the binding energy of [Ni(OH)(H$_2$O)$_3$]$^+$ molecule with SiO$_2$ (−64.82 kcal mol$^{-1}$) is significantly higher than that of with PMO (−56.01 kcal mol$^{-1}$) (Supplementary Data 2, Fig. 4g), thereby inducing the selective growth of hydrated metal compound on SiO$_2$ surface.

Moreover, the electrostatic potential distribution of [Y(H$_2$O)$_6$]$^{3+}$, a representative model of trivalent hydrated cations, on the surface of silica and ethyl-bridged organosilica molecules is calculated (Supplementary Fig. 25a), respectively. Similar to the case of [Ni(H$_2$O)$_4$]$^{2+}$, the exposed hydrogen atoms of the [Y(H$_2$O)$_6$]$^{3+}$ molecules absorbed on the silica surface have higher peak electrostatic potential energy (-11.9 eV) than that of absorbed on the organosilica surface (-11.4 eV). And the binding energy of [Y(OH)(H$_2$O)$_5$]$^{2+}$ with SiO$_2$ (−89.00 kcal mol$^{-1}$) is also higher than that of with PMO (−83.82 kcal mol$^{-1}$) (Supplementary Data 3 and Supplementary Fig. 25b), which contributes to the selective growth of hydrated metal compound on SiO$_2$ surface.

**Fe-sSiO$_2$&rPMO-GOx mJNPs for efficient chemodynamic therapy**
Through rational design, M-mJNPs can integrate the abundant functionalities of metal compounds and the structural advantages of Janus structure to improve their application performance. As a

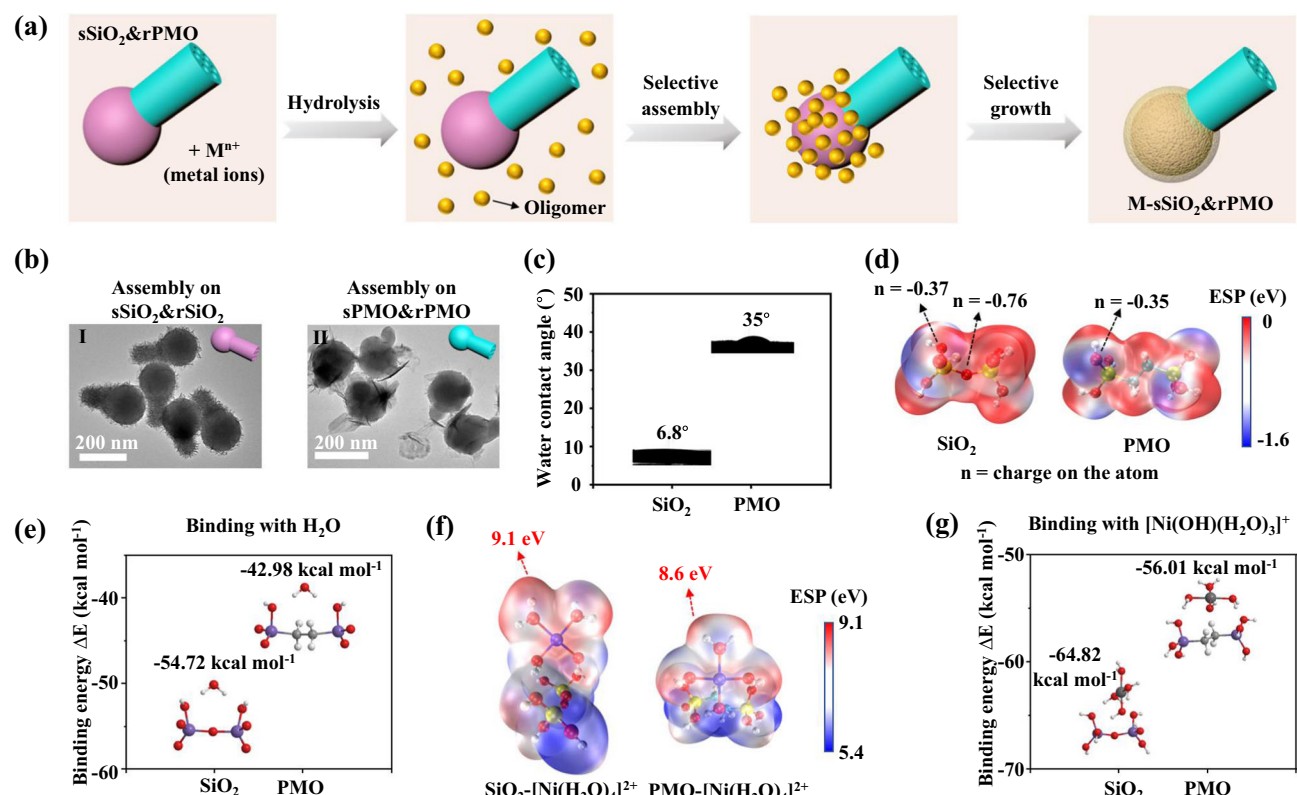

**Fig. 4 | The mechanism of the hydrophilicity-mediated interfacial selective growth strategy. a** Scheme illustration of the selective assembly of the hydrolyzed metal precursors on the hydrophilic $SiO_2$. **b** TEM images of the products synthesized by conducting the same assembly process on $sSiO_2$&$rSiO_2$ and $sPMO$&$rPMO$ with isotropic surface properties. **c** Contact angles of water on the $SiO_2$ and PMO. **d** The electrostatic potential mapping of $SiO_2$ and PMO (ethyl-bridged organosilica) and the charge numbers on the atoms. **e** The interaction energies of water molecule on $SiO_2$ and PMO calculated via DFT simulation. **f** The electrostatic potential mappings of $SiO_2$-$[Ni(H_2O)_4]^{2+}$ and PMO-$[Ni(H_2O)_4]^{2+}$ composites. **g** The interaction energies of $[Ni(OH)(H_2O)_3]^+$ on $SiO_2$ and PMO.

proof-of-concept, we designed a spatially asymmetric cascade nanocatalyst based on the Fe-$sSiO_2$&rPMO-GOx mJNPs for efficient chemodynamic therapy (CDT)[43,44] (Fig. 5a). In this nanocatalyst, the Fe-based nanosheets composed of mixed FeO and $Fe_2O_3$ are selectively coated on $sSiO_2$ head to form the Fe-$sSiO_2$ functional subunit (~120 nm), and rod-shaped rPMO tail (~200 nm in length and ~100 nm in width) were modified with glucose oxidase (GOx) enzyme to form the rPMO-GOx functional subunit (Fig. 5b, Supplementary Figs. 26–30). In this rationally designed Janus nanocatalyst, the rPMO-GOx subunit can specifically oxidize β-D-glucose into gluconic acid and $H_2O_2$, then $H_2O_2$ was subsequently catalyzed by $Fe^{2+}$/$Fe^{3+}$ ions in the Fe-$sSiO_2$ subunit to generate high-toxic •OH via the Fenton reaction[45–47].

Compared with the commonly used spatially indistinguishable combinations of GOx natural enzymes and Fe-based Fenton agent, the spatial isolation of rPMO-GOx and Fe-$sSiO_2$ functional subunits in the rationally designed mJNPs can suppress the adverse effect of strong oxidizing •OH on the activity of GOx, and avoid GOx coverage of the active sites on the surface of Fenton agent. Fe-$sSiO_2$ nanoparticles were synthesized by using the $SiO_2$ nanospheres as templates, and further grafted with GOx to serve as the control sample (Fe-$sSiO_2$-GOx) (Supplementary Figs. 31–34). In order to make the two samples comparable in subsequent experiments, the relative amounts of GOx in two samples were kept nearly the same (compared to the Fe amount), which was about 38.57% for Fe-$sSiO_2$&rPMO-GOx and 33.55% for Fe-$sSiO_2$-GOx (Supplementary Fig. 35). The GOx activity was evaluated by using GOx Activity Assay Kit, it is found that the GOx retained ~62% of its initial activity after the EDS-NHS coupling.

The catalysis performance for •OH generation was evaluated by a typical colorimetric method based on the oxidation of colorless 3,3′,5,5′-tetramethylbenzidine (TMB) to blue oxTMB. After the addition of Fe-$sSiO_2$&rPMO-GOx mJNPs into TMB-glucose solution, the color of solution changed to blue with a strong absorption at 650 nm (Fig. 5c, Supplementary Fig. 36) quickly, confirming the production of •OH. With the increase of glucose concentration, the absorption intensity at 650 nm increased. Impressively, under the catalysis of Fe-$sSiO_2$&rPMO-GOx mJNPs, the blue color of solution is darker than that of the solution catalyzed by Fe-$sSiO_2$-GOx at the same glucose concentration (Fig. 5d, Supplementary Fig. 37).

The time-course absorbance upon the addition of Fe-$sSiO_2$&rPMO-GOx mJNPs into TMB-glucose mixture solution at different glucose concentrations was measured and plotted in Fig. 5e. Induction period, which is often observed in multistep catalytic reactions[48,49], appeared in this cascade catalysis. Longer diffusion pathway of $H_2O_2$ on Fe-$sSiO_2$&rPMO-GOx mJNPs plays an important role in producing induction period in this cascade reaction. The induction periods are more obvious in the catalytic reaction on asymmetric Fe-$sSiO_2$&PMO-GOx mJNPs than that of on symmetry Fe-$sSiO_2$-GOx. The corresponding absorbance-changing rate could be calculated and further converted into initial velocities of •OH generation ($v_0$) via the Beer-Lambert law. Based on the initial velocities ($v_0$) of the cascade catalytic reactions at different glucose concentration ($[s]$), we further calculated maximum velocity ($V_{max}$) and Michaelis-Menten constant ($K_m$) according to Michaelis-Menten equation (Eq. 2) to evaluate the catalytic activities of the two nanocomposites with different architectures.

$$v_0 = \frac{V_{max} * [s]}{K_m + [s]} \tag{2}$$

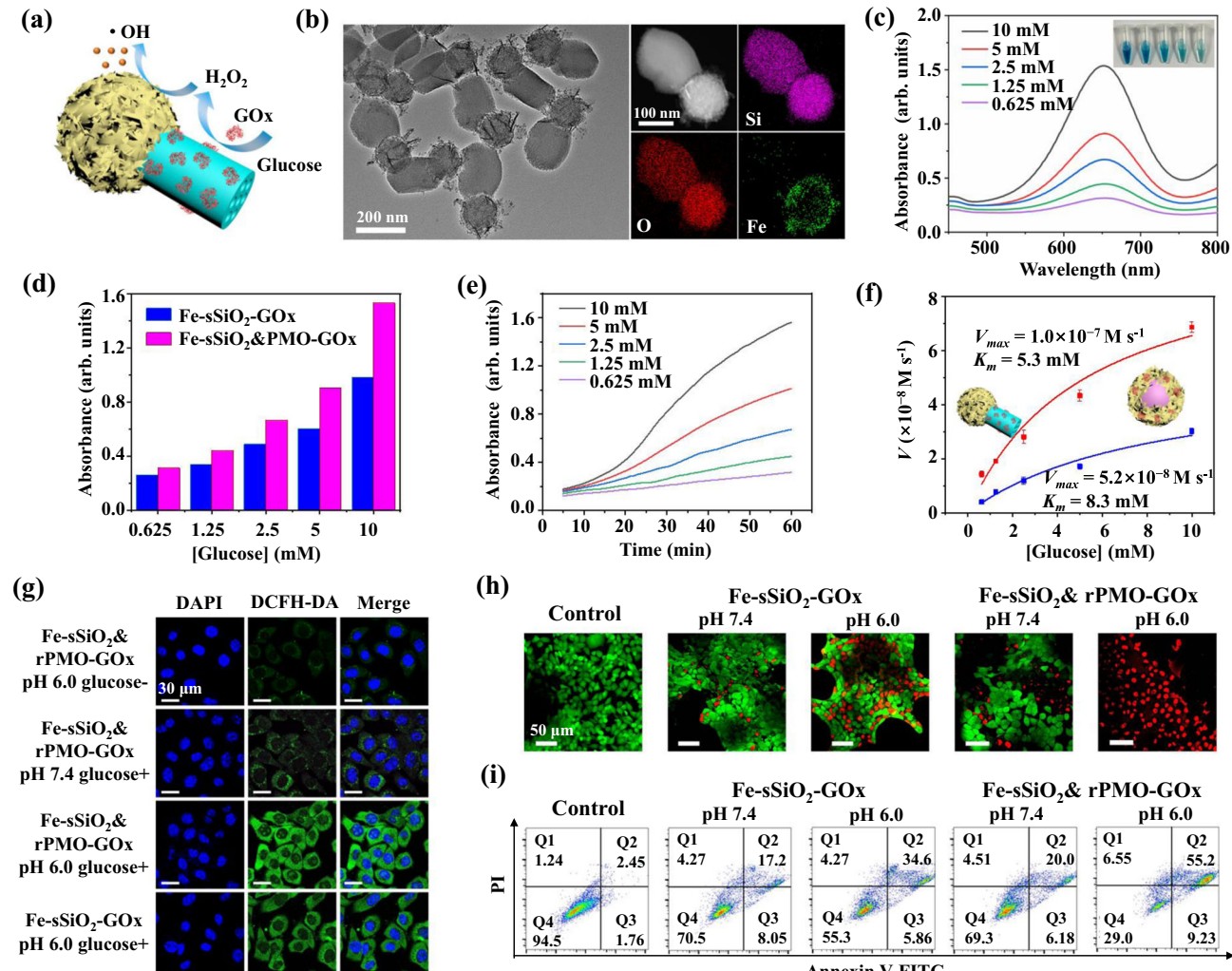

**Fig. 5 | Spatially asymmetric cascade nanocatalyst for enhanced CDT. a** Schematic illustration of cascade nanocatalysts based on Fe-sSiO₂&rPMO-GOx mJNPs. **b** TEM image, HAADF-STEM and EDS elemental mapping images of Fe-sSiO₂&rPMO mJNPs. **c** UV–vis absorption spectra of TMB solutions under catalysis of Fe-sSiO₂&rPMO-GOx mJNPs (Fe concentration: 10 μg mL⁻¹) upon the addition of varied concentrations of β-D-glucose for 1 h. **d** The intensity of the absorption peak of TMB solutions at 650 nm under catalysis of Fe-sSiO₂-GOx or Fe-sSiO₂&PMO-GOx upon the addition of varied concentrations of β-D-glucose for 1 h. **e** Time-dependent absorbance changes at 650 nm of TMB solutions under the catalysis of Fe-sSiO₂&rPMO-GOx mJNPs at different β-D-glucose concentrations. **f** Michaelis–Menten kinetics analysis of the cascade catalytic reaction under the catalysis of Fe-sSiO₂&rPMO-GOx mJNPs or Fe-sSiO₂-GOx. Error bars represent means ± SD from three independent experiments. **g** CLSM images of ROS probe in the HepG2 cell after different treatments. The cells were co-incubated with the Fe-sSiO₂&rPMO-GOx or Fe-sSiO₂-GOx in the presence and absence of glucose (5 mM) at pH 7.4 and 6.0 for 4 h. **h** CLSM images of viable and dead HepG2 cells (stained with calcein-AM/PI) after different treatments in the presence of glucose (5 mM). **i** Flow cytometric quantitative analyses of Annexin V-FITC/PI co-stained HepG2 cells after co-incubation with sSiO₂&rPMO-GOx or Fe-sSiO₂-GOx nanocatalysts under neutral and acidic conditions in the presence of glucose (5 mM). Source data are provided as a Source Data file.

The $K_m$ and $V_{max}$ values were calculated to be 5.3 mM and $1.0 \times 10^{-7}$ M s⁻¹ for Fe-sSiO₂&rPMO-GOx mJNPs, which is much better than that of the Fe-SiO₂-GOx nanocomposite with $K_m$ and $V_{max}$ of 8.3 mM and $5.2 \times 10^{-8}$ M s⁻¹, respectively (Fig. 5f), indicating the superiority of Janus structure in cascade catalytic reactions.

We then evaluated the potential of this Fe-sSiO₂&rPMO-GOx mJNPs in CDT. Both the Fe-sSiO₂&rPMO and Fe-sSiO₂ can be efficiently endocytosed by HepG2 cells with similar endocytosis amount (Supplementary Fig. 38), indicating that the effect of morphology on endocytosis efficiency can be excluded. 2,7-Dichlorofluorescein diacetate (DCFH-DA) probe is introduced to detect the intracellular •OH generation. With the absence of glucose or under neutral condition, the green fluorescence generated from the oxidized DCFH are very weak in cells co-incubated with the Janus Fe-sSiO₂&rPMO-GOx or Fe-sSiO₂-GOx nanocatalysts (Fig. 5g, Supplementary Fig. 39). After co-incubation with Janus Fe-sSiO₂&rPMO-GOx in the glucose solution

(5 mM) under a mild acid condition (pH 6.0), the green fluorescence in HepG2 cells can be clearly observed, which is brighter than that of in the Fe-sSiO₂-GOx group (Fig. 5g), indicating the higher intracellular ROS concentration in Fe-sSiO₂&rPMO-GOx treated group. Evaluated by cell-counting kit-8 (CCK-8) assay, the nanocatalysts exhibited negligible cytotoxicity to human umbilical vein endothelial cells (HUVEC), indicating a good biocompatibility of the nanocatalysts for normal cells (Supplementary Fig. 40). However, for the HepG2 cancer cells, the cell viabilities decrease as the increase of nanocatalysts' concentration (Supplementary Fig. 41). After co-incubation with Janus Fe-sSiO₂&rPMO-GOx nanocatalysts (Fe concentration: 5 μg mL⁻¹) in the mild acidic solution (pH 6.0), ~63.5% of HepG2 cells are killed. And the cancer cell killing efficiency of the Fe-sSiO₂&rPMO-GOx nanocatalysts is much higher than that of the Fe-sSiO₂-GOx nanocatalysts, which can be attributed to the higher •OH generation efficiency of the Janus nanocatalysts. The live/dead cell staining results also vividly

demonstrate the remarkable cancer cell killing efficacy of the Janus nanocatalyst under mild acidic condition (Fig. 5h). Furthermore, the cell apoptosis of different treatments in acidic and neutral culture media were quantitatively analyzed by using the annexin V-FITC/PI apoptosis detection kit (Supplementary Fig. 42). The apoptosis rate of HepG2 cells in the Janus Fe-sSiO$_2$&rPMO-GOx group (55.2%) is much higher than that of in the Fe-sSiO$_2$-GOx group (34.6%) at pH 6.0 (Fig. 5i), further demonstrating the enhanced CDT efficacy of the Janus nanostructure.

In summary, we have developed an interfacial selective growth strategy for the versatile synthesis of metal-compound based mJNPs. Starting from the silica-based mJNPs template with anisotropic dual-surface of hydrophilic SiO$_2$ and hydrophobic PMO, metal precursors can selectively deposit onto the hydrophilic SiO$_2$ subunit to form the metal-compound based mJNPs. The morphology of the obtained metal-compound mJNPs can be well tuned by using different pristine templates. In addition, this method shows good universality and can be used for the synthesis of more than 20 kinds of metal-compound mJNPs, including alkali-earth metal compounds, transition metal compounds, rare-earth metal compounds, etc. The compositions of the metal-compound mJNPs can be tuned from single to multiple metal elements, even high-entropy complexes. As a proof of concept, the Fe-sSiO$_2$&rPMO-GOx mJNPs are rationally synthesized and served as spatially asymmetric cascade nanocatalysts, in which the PMO-GOx functional subunits could effectively deplete glucose in tumor cells, and meanwhile produce a considerable amount of H$_2$O$_2$ for subsequent Fenton reaction under the catalysis of Fe-sSiO$_2$ functional subunits in the tumor microenvironment. Taking advantage of the spatial isolation of the dual functional subunits, the cascade catalytic efficiency of the mJNPs nanocatalysts is greatly increased, thus realizing remarkably efficient CDT for cancer cell killing and tumor restrain.

## Methods

### Materials and chemicals

Calcium chloride dihydrate (CaCl$_2$·2H$_2$O, ≥99%), magnesium nitrate hexahydrate (Mg(NO$_3$)$_2$·6H$_2$O, ≥98%) manganese chloride tetrahydrate (MnCl$_2$·4H$_2$O, 99.99%), cobalt nitrate hexahydrate (Co(NO$_3$)$_2$·6H$_2$O, 99.99%), nickel chloride hexahydrate (NiCl$_2$·6H$_2$O, 99.99%), copper nitrate hydrate (Cu(NO$_3$)$_2$·3H$_2$O, 99.99%), zinc nitrate hexahydrate (Zn(NO$_3$)$_2$·6H$_2$O, 99.99%), iron sulfate heptahydrate (FeSO$_4$·7H$_2$O, 99.95%), yttrium (III) chloride hexahydrate (YCl$_3$·6H$_2$O, 99.9%), ytterbium (III) chloride hexahydrate (YbCl$_3$, 99.9%), gadolinium (III) chloride hexahydrate (GdCl$_3$·6H$_2$O, 99.9%), erbium (III) chloride anhydrous (ErCl$_3$, 99.9%), neodymium (III) chloride hexahydrate (NdCl$_3$·6H$_2$O, 99.9%), dysprosium (III) chloride hexahydrate (DyCl$_3$·6H$_2$O, 99.9%), lutetium (III) chloride hexahydrate (LuCl$_3$·6H$_2$O, 99.9%), thulium (III) chloride hexahydrate (TmCl$_3$·6H$_2$O, 99.9%), holmium (III) chloride hexahydrate (HoCl$_3$·6H$_2$O, 99.9%), lanthanum (III) chloride hydrate (LaCl$_3$·xH$_2$O, 99.9%), praseodymium (III) chloride hydrate (PrCl$_3$·xH$_2$O, 99.9%), terbium(III) chloride hexahydrate(TbCl$_3$·6H$_2$O, 99.9%), tetraethyl orthosilicate (TEOS, 99%), hexamethylenetetramine (HMTA), hydrogen peroxide (H$_2$O$_2$, 30 wt%), glucose oxidase (GOx), (3-aminopropyl)triethoxysilane (APTES, 99%), poly(diallyl di methyl ammonium chloride) solution (PDDA, 20 wt. % in water), 1-ethyl-3-(3-dimethylaminopropyl) carbodiimide hydrochloride (EDC), N-hydroxysuccinimide (NHS), 3,3',5,5'-tetramethylbenzidine (TMB), bovine serum albumin (BSA > 98%) and fluorescein5(6)-isothiocyanate (FITC) were purchased from Aladdin Reagent Co., Ltd. Hexadecyltrimethylammonium bromide (CTAB, 99%) and bis(triethoxysilyl)ethane (BTEE, 96%) were purchased from Sigma-Aldrich. Sodium hydroxide (NaOH, >96%), cyclohexane (AR), ethanol (AR), ammonium nitrate (NH$_4$NO$_3$, 98%) and ammonium hydroxide solution (28 wt.% NH$_3$ in H$_2$O) were purchased from Shanghai Chemical Reagents Co., Ltd. Phosphate buffer saline (PBS)

solution, 4′,6-diamidino-2-phenylindole (DAPI), Cell Counting Kit-8 (CCK-8), Calcein-AM/propidium iodide (PI) and Reactive Oxygen Species Assay Kit were purchased from Bestbio Biotech (China). Glucose Oxidase Activity Assay Kit purchased from Beijing Solarbio Science & Technology Co., Ltd. Deionized water was used throughout the experiments.

### Synthesis of silica-based mesoporous Janus templates

Four kinds of Janus templates with anisotropic or isotropic surface properties were synthesized as follows.

**Synthesis of silica nanospheres (sSiO$_2$).** The colloidal SiO$_2$ nanospheres were prepared via a modified Stöber method[50]. Typically, 150.0 mg of CTAB was dispersed in 50.0 mL of deionized water and 30.0 mL of ethanol by sonication. Then, 550 μL of ammonia aqueous solution (28 wt%) was added under stirring. After stirring for 30 min, 800 μL of TEOS was injected and the reaction continued for 9 h. The sSiO$_2$ nanoparticles were collected by centrifugation at 17,726 × g for 5 min, washed with ethanol and water for several times and dried for further use.

**Synthesis of Janus sSiO$_2$&rPMO.** The Janus sSiO$_2$&rPMO nanocomposites (rPMO: rod-like periodic mesoporous organosilica) were prepared based on the anisotropic growth method[36]. Typically, 8.0 mg of the sSiO$_2$ powder obtained above was added into the mixture containing 57.0 mL of water, 3.0 mL of ethanol and 225.0 mg of CTAB. After sonification for 30 min, 3.0 mL of ammonia aqueous solution (28 wt.%) was added into the mixture under continuous stirring. After stirring for 30 min, 75 μL of BTEE was added into the reaction solution, and the reaction was continued for 2 h. The Janus sSiO$_2$&rPMO nanocomposites were collected by centrifugation at 17,726 × g for 5 min and washed with ethanol for several times.

**Synthesis of Janus sSiO$_2$&rSiO$_2$.** The Janus sSiO$_2$&rSiO$_2$ nanocomposites were fabricated via the anisotropic growth strategy[41]. Typically, 8.0 mg of the sSiO$_2$ nanoparticles obtained above was dispersed in 20.0 mL of water with 100.0 mg of CTAB, and the mixture was sonicated for 30 min. Then, 1.0 mL of ammonia aqueous solution (28 wt.%) was added into the mixture under continuous stirring. After 30 min stirring, 80 μL of TEOS were added. The reaction was allowed to proceed for 6 h. The sample was collected by centrifugation at 17,726 × g for 5 min and washed with water and ethanol for several times.

**Synthesis of spherical PMO nanoparticles (sPMO).** The sPMO nanospheres were synthesized via a modified Stöber method[50]. Typically, 80.0 mg of CTAB were dispersed in a mixed solution containing 28.0 mL of deionized water and 12.0 mL of ethanol. Then 400 μL of ammonia aqueous solution (28 wt.%) was added under continuous stirring. After stirring for 30 min, 100 μL of BTEE was added into the dispersion. The reaction was allowed to react for 12 h. Then, the products were collected by centrifugation at 17,726 × g for 5 min and washed with water and ethanol for several times.

**Synthesis of Janus sPMO&rSiO$_2$.** The synthesis process of sPMO&rSiO$_2$ is similar to that of sSiO$_2$&rSiO$_2$, which can be obtained by substituting 10.0 mg of sPMO nanoparticles for 8.0 mg of sSiO$_2$ nanoparticles in the same reaction system.

**Synthesis of Janus sPMO&rPMO.** The synthesis process of sPMO&rPMO is similar to that of sSiO$_2$&rPMO, which can be obtained by substituting 10.0 mg of sPMO nanoparticles for 8.0 mg of sSiO$_2$ nanoparticles in the same reaction system.

**Synthesis of rod-like silica nanoparticles (rSiO$_2$).** 100.0 mg of CTAB was dispersed in 50.0 mL of deionized water and 5.0 mL of ethanol by sonication. Then, 1.5 mL of ammonia aqueous solution (28 wt%) was added under stirring. After stirring for 30 min, 600 µL of TEOS was injected and the reaction continued for 9 h. The rSiO$_2$ nanoparticles were collected by centrifugation at 17,726 × g for 5 min, washed with ethanol and water for several times and dried for further use.

**Synthesis of rod-like PMO (rPMO).** 150.0 mg of CTAB was dispersed in 50.0 mL of deionized water and 3.5 mL of ethanol by sonication. Then, 1.5 mL of ammonia aqueous solution (28 wt%) was added under stirring. After stirring for 30 min, 80 µL of BTEE was injected and the reaction continued for 9 h. The rPMO nanoparticles were collected by centrifugation at 17,726 × g for 5 min, washed with ethanol and water for several times and dried for further use.

In order to remove the CTAB surfactant, all the obtained samples went through an extraction process, during which the sample was extracted twice with an alcoholic solution of NH$_4$NO$_3$ (6 g L$^{-1}$) at 60 °C and washed three times with ethanol and deionized water.

**Synthesis of metal-compounds based mJNPs with single metal element (M-sSiO$_2$&rPMO)**
Generally, for a synthesis of M-sSiO$_2$&rPMO mJNPs, a certain amount of metal precursor was added into 30.0 mL of Janus sSiO$_2$&rPMO aqueous solution (1.0 mg mL$^{-1}$). After stirring for 30 min, a certain amount of HMTA was added into the mixture to catalyze the hydrolysis of the metal precursor. The specific experimental conditions for the synthesis of different M-sSiO$_2$&rPMO mJNPs are as follows.

**Mg-, Mn-, Co, Ni, Cd and all the rare-earth based M-sSiO$_2$&rPMO mJNPs.** 200 µL of 0.2 M metal precursor solution and 3750 µL of 0.2 M HMTA solution was added to the reaction system subsequentially with an interval of 30 min. The reaction was reacted at 90 °C for 30 min.

**Cu-sSiO$_2$&rPMO mJNPs.** 200 µL of 0.2 M Cu(NO$_3$)$_2$ solution and 60 µL of 0.2 M HMTA solution were added to the reaction system subsequentially with an interval of 30 min. The reaction was reacted at 85 °C for 10 h.

**Zn-sSiO$_2$&rPMO mJNPs.** 200 µL of 0.2 M Zn(NO$_3$)$_2$ solution and 600 µL of 0.2 M HMTA solution were added to the reaction system subsequentially with an interval of 30 min. The reaction was reacted at 90 °C for 30 min.

To transform the above M-sSiO$_2$&rPMO mJNPs into metal oxides based mJNPs, the obtained samples were further calcinated at 600 °C for 6 h.

**Fe-sSiO$_2$&rPMO mJNPs.** Under nitrogen atmosphere, 200 µL of 0.2 M FeSO$_4$ solution and 3750 µL of 0.2 M HMTA solution was added into the reaction system subsequentially with an interval of 30 min. The react solution was maintained at 90 °C for 30 min. Then, the reaction was switch to air atmosphere and reacted for another 2 h.

**Ca-sSiO$_2$&rPMO mJNPs.** 150.0 mg of CaCl$_2$·2H$_2$O powder, and 80.0 mg of Janus sSiO$_2$&rPMO were added in a flat bottom beaker containing 80.0 mL of ethanol. Then the flat bottom beaker was covered by a plastic wrap which was stabbed with several pores. 5000.0 mg of NH$_4$HCO$_3$ was added in another flat bottom beaker. Then, the two flat bottom beakers were placed in a sealed container. The reaction was performed at 40 °C for 60 h and the obtained Ca-sSiO$_2$&rPMO was collected by centrifugation at 177,26 × g for 5 min and washed with ethanol for several times.

**Synthesis of metal-compounds based mJNPs with multiple components (M$_x$-sSiO$_2$&rPMO)**
For M$_x$-sSiO$_2$&rPMO mJNPs, 200 µL of precursors solution containing multiple metal salts was added into 30.0 mL of Janus sSiO$_2$&rPMO solution (1 mg mL$^{-1}$) and stirred for 30 min. Then, 3750 µL of HMTA solution (0.2 M) was added into the mixture, and the reaction was reacted at 90 °C for 30 min to obtain the M$_x$-sSiO$_2$&rPMO mJNPs.

**Mn/Ni-sSiO$_2$&rPMO mJNPs.** The precursors solution is prepared by simply mixing equal amount of 0.2 M MnCl$_2$ solution and 0.2 M Ni(NO$_3$)$_2$ solution.

**Co/Ni-sSiO$_2$&rPMO mJNPs.** The precursors solution is prepared by simply mixing equal amount of 0.2 M Co(NO$_3$)$_2$ solution and 0.2 M Ni(NO$_3$)$_2$ solution.

**Gd/Yb-sSiO$_2$&rPMO mJNPs.** The precursors solution is prepared by simply mixing equal amount of 0.2 M GdCl$_3$ solution and 0.2 M YbCl$_3$ solution.

**Mn/Co/Ni-sSiO$_2$&rPMO mJNPs.** The precursors solution is prepared by simply mixing equal amount of 0.2 M MnCl$_2$ solution, 0.2 M Co(NO$_3$)$_2$ solution and 0.2 M Ni(NO$_3$)$_2$ solution.

**Y/Yb/Gd-sSiO$_2$&rPMO mJNPs.** The precursors solution is prepared by simply mixing equal amount of 0.2 M YCl$_3$ solution, 0.2 M YbCl$_3$ solution and 0.2 M GdCl$_3$ solution.

**Tm/Y/La/Pr/Tb/Nd/Ho/Dy/Lu-sSiO$_2$&rPMO mJNPs.** The precursors solution is prepared by mixing equal amount of 0.2 M TmCl$_3$ solution, 0.2 M YCl$_3$ solution, 0.2 M LaCl$_3$ solution, 0.2 M PrCl$_3$ solution, 0.2 M TbCl$_3$ solution, 0.2 M NdCl$_3$ solution, 0.2 M HoCl$_3$ solution, 0.2 M DyCl$_3$ solution and 0.2 M LuCl$_3$ solution.

**Tb/Ho/Tm/Dy/Er/Pr/Nd/Lu-sSiO$_2$&rPMO mJNPs.** The precursors solution is prepared by mixing equal amount of 0.2 M TbCl$_3$ solution, 0.2 M HoCl$_3$ solution, 0.2 M TmCl$_3$ solution, 0.2 M DyCl$_3$ solution, 0.2 M ErCl$_3$ solution, 0.2 M PrCl$_3$ solution, 0.2 M NdCl$_3$ solution and 0.2 M LuCl$_3$ solution.

To transform the above M-sSiO$_2$&rPMO into metal oxides based mJNPs, the obtained samples were further calcinated at 700 °C for 6 h.

**Grafting GOx on Fe-SiO$_2$&PMO and Fe-SiO$_2$**
The GOx was modified to the Fe-SiO$_2$&PMO mJNP and Fe-SiO$_2$ nanoparticles through the following two steps.

**Preparation of amino-functionalized Fe-SiO$_2$&PMO and Fe-SiO$_2$ nanoparticles.** The obtained Fe-SiO$_2$&PMO and Fe-SiO$_2$ nanoparticles were functionalized with amino group via a post-grafting strategy. For Fe-SiO$_2$&PMO-NH$_2$, 200 µL of APTES was added into 50.0 mL of Fe-SiO$_2$&PMO ethanol suspension (1.0 mg mL$^{-1}$). The mixture was stirred at 60 °C for 12 h. The products were collected by centrifugation and washed with water and ethanol for several times. For Fe-SiO$_2$-NH$_2$, 10.0 mg of PDDA was added into 50.0 mL of Fe-SiO$_2$ ethanol suspension (1.0 mg mL$^{-1}$) and stirred for 12 h. The products were collected by centrifugation at 17,726 × g for 5 min and washed with water and ethanol for several times.

**Preparation of Fe-SiO$_2$&PMO-GOx and Fe-SiO$_2$-GOx nanoparticles.** 60.0 mg of EDC and 80.0 mg of NHS were dissolved into 10.0 mL of PBS (pH 6.0). After the addition of GOx (2.0 mg), the mixture was allowed to react for 2 h at room temperature. Then, the Fe-SiO$_2$&PMO-NH$_2$ or Fe-SiO$_2$-NH$_2$ was added to the activated GOx solution. After reacting for 4 h under stirring, the obtained Fe-SiO$_2$&PMO-GOx or

Fe-SiO$_2$-GOx nanocomposites were collected by centrifugation at 9690 × $g$ for 5 min.

## Synthesis of FITC labeled nanoparticles

Amino-functionalized nanoparticles of Fe-SiO$_2$&PMO-NH$_2$ or Fe-SiO$_2$-NH$_2$ (20.0 mg) were dispersed in 10.0 mL ethanol containing 2.0 mg FITC under dark condition. After stirring (500 rpm) at room temperature overnight, the obtained nanoparticles were washed by water and ethanol for several times and dried in vacuum at 40 °C for 12 h for further use.

## Detection of hydroxyl radicals in the solution

TMB was used as probe to evaluate the generation of •OH in PBS solutions (pH 6.0). With the presence of glucose, the cascade catalysts can sequentially catalyze the generation of •OH with strong oxidation properties, which further result in the oxidation of colorless TMB to blue oxTMB with characteristic absorption peak at 650 nm. The concentration of the catalyst and TMB were 10 μg mL$^{-1}$ (Fe concentration) and 2 mM, respectively.

## In vitro cytotoxicity evaluation

Human umbilical vein endothelial cell (HUVEC) and HepG2 cell were purchased from cell bank of Chinese academy of science (Shanghai, China). The cells were cultured in standard Roswell Park Memorial Institute (RPMI) 1640 medium supplemented with 10% (v/v) FBS, 100 mg mL$^{-1}$ streptomycin and 100 U mL$^{-1}$ penicillin at 37 °C in a humidified incubator with 5% CO$_2$. The cytotoxicity of the prepared Fe-SiO$_2$&PMO-GOx and Fe-SiO$_2$-GOx nanocomposites was assessed using standard Cell Counting Kit-8 (CCK-8) assay. HUVEC or HepG2 cells were seeded in 96-well plates (10$^4$ cells per well) for 24 h. After that, the cells were incubated with fresh medium containing different concentrations of Fe-SiO$_2$&PMO-GOx and Fe-SiO$_2$-GOx nanocatalysts (Fe concentration: 0.313, 0.625, 1.25, 2.5, 5, 10 μg mL$^{-1}$) for another 24 h. Then, the medium was discarded and the cells were washed with PBS. The mixture of CCK-8 and fresh culture medium was added into each well and incubated for 2 h. Finally, the cell viability was evaluated by measuring the absorbance at the wavelength of 450 nm. The following formula was used to calculate the relative cell viability: Relative viability (%) = (mean of absorbance value of treatment group/mean absorbance value of control) × 100%. Cells treated with pure medium without the nanocomposites was used as control group. Four parallel experiments were performed for each group.

## In vitro cell uptake of the nanocatalysts

The intracellular endocytosis of Fe-sSiO$_2$&rPMO and Fe-sSiO$_2$ were investigated by confocal laser scanning microscopy (CLSM). HepG2 cells were seeded in the CLSM-exclusive culture dishes (10$^5$ cells per dish) and incubated for 24 h. Then, the culture media were replaced by Fe-sSiO$_2$&rPMO-FITC or Fe-sSiO$_2$-FITC (dispersed in pH 7.4 or 6.0 culture medium, Fe concentration 5 μg mL$^{-1}$). After co-incubation for 6 h, the HepG2 cells were washed with PBS and stained with DAPI and imaged by CLSM.

## In vitro ROS detection

CLSM were introduced to evaluate the in vitro ROS generation ability of cascade nanocatalysts at different pH values. For CLSM observation, HepG2 cells were seeded in the CLSM-exclusive culture dishes (10$^5$ cells per dish) and incubated for 24 h. The cells were then co-incubated with Fe-sSiO$_2$&rPMO or Fe-sSiO$_2$ (dispersed in pH 7.4 or 6.0 culture medium, Fe concentration: 5 μg mL$^{-1}$) in the RPMI 1640 culture medium for 6 h. Then the cells were washed with PBS and further cultured in fresh medium containing DCFH-DA probe for 30 min for the CLSM observation.

## Evaluation of in vitro therapeutic efficacy

HepG2 cells were seeded on a 96-well plate at a density of 10$^4$ cells per well and incubated for 24 h. After that, the cells were incubated with Fe-sSiO$_2$&rPMO-GOx and Fe-sSiO$_2$-GOx under different pH conditions (7.4 or 6.0) for 6 h. Then, cells were rinsed with PBS for several times and the standard CCK-8 assay was conducted to evaluate the cell viability of each group. For visualizing the killing effect, the cells were also stained with Calcein-AM/PI after different treatments and observed by CLSM.

## Computational details

The density functional theory (DFT) calculation were conducted in Gaussian (G09) program. The structure optimization was performed at PBE0-D3(BJ)/6-311 g* level. The binding energy ($E_B$) between SiO$_2$/PMO and H$_2$O/[Ni(OH)(H$_2$O)$_3$]$^+$ is defined as following: $E_B = E_{Complex} - E_s - E_m$, where $E_{Complex}$ is the total energy of the complex after binding, $E_s$ is the total energy of silica or organosilica molecule, $E_m$ is the total energy of H$_2$O or [Ni(OH)(H$_2$O)$_3$]$^+$. The electrostatic potential (ESP) was analyzed by Multiwfn package and VMD package[51,52].

## Characterization

Transmission electron microscopy (TEM) images were acquired on a Hitachi HT7700 transmission electron microscope operating at 120 kV. High-resolution TEM (HRTEM), high-angle annular dark field imaging in the scanning TEM (HAADF-STEM) and energy-dispersive X-ray spectroscopy (EDS) mapping images were obtained on JEM-2100F microscope (JEOL, Japan) with an accelerating voltage of 200 kV equipped with a post-column Gatan imaging filter. Scanning electron microscopy (SEM) images were captured using field emission scanning electron microscopy (FESEM, Hitachi S-4800, Japan). X-ray diffraction (XRD) patterns were collected on with a Bruker D8 powder X-ray diffractometer (Germany) using Cu-K$_α$ radiation (40 kV, 40 mA). The obtained XRD data was analyzed by using Highscore 4.0. The contents of Fe in Fe-sSiO$_2$&rPMO mJNPs and Fe-sSiO$_2$ nanoparticles were determined by inductively coupled plasma-optical emission spectroscopy (ICP-OES) system (Varian 710-ES). Nitrogen adsorption–desorption measurements were conducted to obtain information on the porosity. The measurements were conducted at 77 K with ASAP 2420. Before measurements, the samples were degassed in vacuum at 180 °C for at least 12 h. The Brunauer-Emmett-Teller (BET) method was utilized to calculate the specific surface areas and the Barrett-Joyner-Halenda (BJH) model was utilized to calculate the pore volumes and the pore size distributions derived from the adsorption branches of isotherms. XPS characterization was performed on a Thermo scientific K-Alpha XPS spectrometer with Al Kα radiations, and with the C 1s peak at 284.8 eV as an internal standard for all the spectra. The obtained XPS data was analyzed by using Thermo Advantage 5.99. Size distribution and zeta potential of the samples were recorded by using Zetasizer Nano ZS apparatus (Malvern, UK). Fourier transform infrared (FTIR) spectra were recorded using Fourier transform infrared spectrometer (ThermoFisher, Nicolet iS10, USA). UV–vis absorption spectra were measured on an Epoch-Microplate Spectrophotometer (BioTek, USA). Confocal laser fluorescence microscope (CLSM) images were obtained in IX81, Olympus, Japan. For flow cytometry analysis, cell lines were seeded at densities and in plates prior to the experiment. After incubation with the material of interest, they were washed in PBS and detached with TrypLE Express Enzyme for 5 min at room temperature. The cells were pelleted by spinning at 350 g for 5 min and resuspended in PBS. The collected cells were stained by Annexin V-FITC/PI, finally, the flow cytometry analysis was performed on an Accuri C6 flow cytometer (BD Biosciences, USA) and the percentage of apoptosis was analyzed by using Flowjo 10.6.2.

## Statistics and reproducibility

All the synthetic experiments were repeated at least three times, and similar results were obtained with each batch (Figs. 1b, c, d, f, g, j, k, l; 2a–i; 3b, d, f, h; 4 b; 5b and Supplementary Figs. 1b, c; 2; 5; 7a, b; 8; 9a–l; 10a, b; 11a, b; 17a–f; 19a–c; 20a, b, c, e; 21a, c; 22a; 23a, b; 24b, c; 30; 31a, b). All the cellular experiments characterized by CLSM were repeated at least two times, yielding comparable results (Fig. 5g, h and Supplementary Figs. 38; 39). The sample size in this study was not pre-determined using statistical methods. The experiments were not randomized.

## Reporting summary

Further information on research design is available in the Nature Portfolio Reporting Summary linked to this article.

## Data availability

All the data generated in this study are provided in the main text and Supplementary Information. Source data are provided with this paper.

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

## Acknowledgements

We gratefully acknowledge funding from the National Natural Science Foundation of China (22075049, X.L.; 21875043, X.L.; 22088101, D.Z.; 21701027, X.L.; 21733003, D.Z.; 21905052, W.W.; 51961145403, F.Z.), National Key R&D Program of China (2018YFA0209401, D.Z.), Fundamental Research Funds for the Central Universities (20720220010, X.L.), Key Basic Research Program of Science and Technology Commission of Shanghai Municipality (22JC1410200, D.Z.), Natural Science Foundation of Shanghai (22ZR1478900, X.L.; 18ZR1404600, X.L.; 20490710600, X.L.), Shanghai Rising-Star Program (20QA1401200, X.L.), Shanghai Pilot Program for Basic Research-Fudan University (22TQ004, X.L.).

## Author contributions

X.L., D.Z., F.Z., W.W., and Y.Y. conceived and supervised the research. X.L., W.W., and Y.Y. designed the experiments and performed most of the experiments and data analysis. R.L. performed the DFT calculations. M.L., H.Y., and Y.Y. performed the bio-experiments. E.X. participated the synthesis experiments. X.L. and Y.Y. wrote the paper. All authors discussed the results and commented on the manuscript.

## Competing interests

The authors declare no competing interests.
