## [Peer Review File · Nature Communications]

Reviewer #1 (Remarks to the Author):

The authors reported a method of synthesizing metallic-compound-based mesoporous Janus nanocomposite based on the interfacial selective assembly strategy. Furthermore, the authors demonstrated that such a strategy is hydrophilicity-driven through experimental results and theoretical calculations. A library of metallic-silica Janus nanoparticles has been synthesized and characterized, with a proof-of-concept demonstration of Fe-GOx functionalized Janus nanocomposites for chemodynamic therapy applications.

The reported work is of particular importance to the material science fields, established on the method of anisotropic Janus nanoparticles synthesis based on silica materials. However, such anisotropic synthesis has been extensively studied and published in the literature, and the proof-of-concept functionalization of the materials as a drug-delivery system has also been reported previously. Nevertheless, the manuscript retained its novelty from the systematic study of the metallic-compound-based mJNPs point of view and exhibited sufficient novelty to serve the scientific community. There are some data analyses and conclusions made in the manuscript unpersuasive and should be carefully examined or supported by further analysis. In this case, this manuscript could be published after revision.

1. Page 5, lines 113-114: the dimension of the rod-shaped tail should be reported in length x diameter. It is not obvious if the 120 nm is the length of the rod or the diameter.
2. Page 5, line 131: It would be useful to supply the original BET data of the sSiO₂&rPMO before metallic compound incorporation to compare the porous sutural changes before and after metallic functionalization. It also should be further demonstrated the claim that the metallic compounds are located on the surface of the sSiO₂ ("wrapped by a layer of nanosheet to form a core-shell structure head" line 124) and show that either there is any deposition of the metallic compounds inside the mesoporous structure. It also applies to the case of other metal compounds.
3. Since the metallic compound incorporation is one of the highest novelty of the manuscript, it is highly recommended to supplement the chemical composition analysis by XPS of the metallic compound before and after calcination. For example, if the metallic compound existed in its hydroxide or oxide format (if oxide how is its oxidation state). Also, the amount of metal loading should be included (can be obtained from an elemental percentage from XPS, sometimes, or more precisely from ICP-MS). This is especially true for the proof-of-concept FeO/Fe₂O₃ case. XRD and EDS analysis are not convincing enough to conclude the metallic compound composition.
4. Page 10 Figure 4 (b), the caption of "Assembly on PMO&PMO" should be revised to "sPMO&rPMO".
5. Page 11, lines 264-266. The CA of SiO₂ and PMO was measured on a rod-like structure or sphere structure? Is there any difference? It is also generally accepted that a contact angle <90 degrees is hydrophilic. The CA at 35 degrees should be considered quite hydrophilic as well. Thus it is very questionable to conclude hydrophobicity/hydrophilicity-induced selective synthesis.
6. Page 15, lines 363-366: The enzymatic kinetic analysis shows that GOx after being functionalized on the Fe-mJNPs retained Km in the value of 5.3 mM and 8.3 mM, respectively. However, the native GOx purchased from the indicated manufacturer indicated a Km value of 19 mM. A lower Km means stronger substrate binding affinity. How can the enhanced substrate binding affinity of the GOx-Fe-mJNPs be explained? Also, the turnover number of Kcat (calculated from Vmax/Km) for Fe-sSiO₂&rPMO-GOx mJNPs maintained a value of 2*10⁻⁵ .s⁻¹, while native GOx has a turnover number in the range of 600 to a few thousand per second. Does that mean the enzyme lost its catalytic activity after the EDS-NHS coupling? https://www.aladdin-e.com/zh_cn/g130084.html
7. SI: Page 12, Figure S28: certain glucose negative control is missing in order to make full compensation with Figure 5:
Fe-sSiO₂-GOx pH7.4 glucose-
Fe-sSiO₂&rPMO-GOx pH 6.0 glucose-
Fe-sSiO₂&rPMO-GOx pH 7.4 glucose-

Reviewer #2 (Remarks to the Author):

The authors reported a novel pathway for the synthesis of metallic-compound-based mesoporous Janus nanoparticles (mJNPs) via hydrophilicity-mediated interfacial selective assembly. It is noteworthy that, using the sSiO₂&rPMO template (where SiO₂ and PMO were spherical and rod-shaped, respectively), the authors were able to selectively deposit metal onto the hydrophilic SiO₂-subunit to form metallic-compound-based mJNPs (M-mJNPs), in which more than 20 metallic compound-based mJNPs were successfully synthesized. This finding provides a facile synthesis of metallic-compound-based mJNPs and contributes to adding more varieties of Janus materials. Moreover, the as-synthesized Fe-sSiO₂&rPMO-GOx mJNPs were active as a catalyst for enhanced efficient chemodynamic therapy (CDT). Overall, this paper is clearly written and well-organized, and the introduction is reasonable, given the premise of the paper. The data were of high quality, and the analysis was performed very carefully, while the discussion was written in a very compelling way. This work has successfully answered the challenges in synthesizing Janus materials, e.g., achieving a well-defined interface between two different regions, precise control over the distribution of the loaded metal oxides, and creating a broad platform for constructing different functions and structures. However, the present reviewer felt inconvenienced by the term "metallic" in the M-mJNPs, as the term is usually used to describe zero-valent metals and not the oxide or hydroxide forms of metals. Therefore, it is suggested to use a more appropriate term to represent the metal oxides grown on the surface of sSiO₂ of mJNPs. On the mechanistic study, computational studies on trivalent cations (e.g., Mn³⁺, Y³⁺, or Co³⁺) can be added to confirm the claim on the electrostatic potential distribution of hydrated metal ions on the surface of SiO₂ or PMO. In the time-dependent absorbance changes of TMB solutions over Fe-sSiO₂&rPMO-GOx mJNPs at different β-D-glucose concentrations (Fig. 5d), an induction period is observed in the initial 20 min for all β-D-glucose concentrations. Thus, it is important to further explain this phenomenon in detail.

Manuscript ID: NCOMMS-23-05200A

Title: "Versatile synthesis of metallic-compound based mesoporous Janus nanoparticles "

Point-to-Point Response to the Referees

Reviewer #1:

The authors reported a method of synthesizing metallic-compound-based mesoporous Janus nanocomposite based on the interfacial selective assembly strategy. Furthermore, the authors demonstrated that such a strategy is hydrophilicity-driven through experimental results and theoretical calculations. A library of metallic-silica Janus nanoparticles has been synthesized and characterized, with a proof-of-concept demonstration of Fe-GOx functionalized Janus nanocomposites for chemodynamic therapy applications.

The reported work is of particular importance to the material science fields, established on the method of anisotropic Janus nanoparticles synthesis based on silica materials. However, such anisotropic synthesis has been extensively studied and published in the literature, and the proof-of-concept functionalization of the materials as a drug-delivery system has also been reported previously. Nevertheless, the manuscript retained its novelty from the systematic study of the metallic-compound-based mJNPs point of view and exhibited sufficient novelty to serve the scientific community. There are some data analyses and conclusions made in the manuscript unpersuasive and should be carefully examined or supported by further analysis. In this case, this manuscript could be published after revision.

Response: We thank the reviewer very much for the positive comments.

Comments 1: Page 5, lines 113-114: the dimension of the rod-shaped tail should be reported in length x diameter. It is not obvious if the 120 nm is the length of the rod or the diameter.

Response: We thank the reviewer very much for the comment. The sSiO₂&rPMO template shown in Supplementary Fig. 1 has a rod-shaped tail with a dimension of ~ 120 nm in length and ~ 110 nm in diameter. According to our previous reports (*Adv. Mater.* **2017**, 29, 1701652), the length of the rod-shaped tail can be well tuned from ~ 50 nm to ~ 2 μm.

Accordingly, we have modified the following sentence in the revised manuscript on page 5: "Transmission electron microscope (TEM) image of the obtained sSiO₂&rPMO template with anisotropic dual-surface of hydrophilic and hydrophobic (Supplementary Fig. 1) shows good dispersity and distinctive asymmetric structure consisting of a spherical sSiO₂ head (~ 120 nm) and a rod-shaped tail (~ 120 nm in length and ~ 110 nm in diameter).

Comments 2: Page 5, line 131: It would be useful to supply the original BET data of the sSiO₂&rPMO before metallic compound incorporation to compare the porous sutural changes before and after metallic functionalization. It also should be further demonstrated the claim that the metallic compounds are located on the surface of the sSiO₂ (“wrapped by a layer of nanosheet to form a core-shell structure head” line 124) and show that either there is any deposition of the metallic compounds inside the mesoporous structure. It also applies to the case of other metal compounds.

Response: Thanks for your suggestion. We have accepted it. In the revised manuscript, the nitrogen sorption isotherms of the pristine sSiO₂&rPMO was provided, and a set of pores at ~3 nm was observed for the pristine sSiO₂&rPMO before metallic compound incorporation (Supplementary Fig. 3). After metallic functionalization, the obtained mJNPs possess dual mesopores at about 3 and 9 nm (Fig. 1e), which can be attributed to the mesopores in the pristine sSiO₂&rPMO template and piled mesopores in the Ni-based subunit of mJNPs. So, we can conclude that the deposition of the metallic compounds did not affect the porous structure of the pristine sSiO₂&rPMO. In addition, we also conducted the nitrogen sorption measurement on three other representative M-mJNPs (Ca-, Mn- and Y-mJNPs) to explore their pore size distributions (Supplementary Fig. 15). It is found that the mesopores at ~3 nm are all well maintained in the obtained M-JNPs, further demonstrating that no metallic compounds deposition inside the mesopore channels. Combing the results of pore size distribution analysis and EDS mapping results, it can be validated that the metallic compounds are located on the surface of the sSiO₂.

Accordingly, we have added Supplementary Fig. 3 and Supplementary Fig. 16 in the revised Supporting Information and the corresponding descriptions in the main text of the revised manuscript on Page 5: “The pore size distribution analysis via nitrogen sorption measurement shows that obtained mJNPs possess dual mesopores at about 3 and 9 nm (Fig. 1e), which can be attributed to the mesopores in the pristine sSiO₂&rPMO template (Supplementary Fig. 3) and piled mesopores in the Ni-based subunit of mJNPs. The well retained 3 nm sized mesopore inherited from the pristine template indicates that there is no metal compounds deposition inside the mesoporous structure in the Ni-sSiO₂&rPMO mJNPs.”, **and on Page 8:** “The EDS mapping images further demonstrate that all the metal elements are homogeneously distributed at the head compartment of the mJNPs. And the pore size distribution analysis confirms well reserved mesoporous structure in the mJNPs (Supplementary Fig. 16).”

Supplementary Fig. 3 (a) N_2 sorption isotherms and (b) the corresponding pore size distribution curve of the pristine $\text{sSiO}_2\&\text{rPMO}$.

Supplementary Fig. 16 N_2 sorption isotherms and the corresponding pore size distribution curves of the obtained (a) $\text{Ca-sSiO}_2\&\text{rPMO}$, (b) $\text{Y-sSiO}_2\&\text{rPMO}$ and (c) $\text{Mn-sSiO}_2\&\text{rPMO}$ after calcination.

Comments 3: Since the metallic compound incorporation is one of the highest novelty

of the manuscript, it is highly recommended to supplement the chemical composition analysis by XPS of the metallic compound before and after calcination. For example, if the metallic compound existed in its hydroxide or oxide format (if oxide how is its oxidation state). Also, the amount of metal loading should be included (can be obtained from an elemental percentage from XPS, sometimes, or more precisely from ICP-MS). This is especially true for the proof-of-concept FeO/Fe₂O₃ case. XRD and EDS analysis are not convincing enough to conclude the metallic compound composition.

Response: Thank you for your suggestion. We have accepted it. In the revised manuscript, the chemical composition of the metallic compounds before and after calcination was evaluated by XPS and ICP-OES.

According to the chemical composition analysis by XPS, we have added the description for the characterization method in the main text at Page 25: “XPS characterization was performed on a Thermo scientific K-Alpha XPS spectrometer with Al K α radiations, and with the C 1 s peak at 284.8 eV as an internal standard for all spectra.”, **and Supplementary Fig. 4, 12-15, 18 and 26 with detailed discussion in the revised Supporting Information:**

Supplementary Fig. 4. Ni 2p XPS spectra of the obtained Ni-sSiO₂&rPMO (a) before and (b) after calcination treatment.

Either before or after calcination, the Ni-mJNPs show distinctive signals corresponded to Ni²⁺ species. Before calcination, the XPS spectrum of Ni-sSiO₂&rPMO exhibits a sharp peak centered at 856.08 eV corresponding to Ni 2p_{3/2} with a satellite at 862.18 eV and a Ni 2p_{1/2} peak centered at 873.68 eV with a satellite at 879.88 eV, which is well matched with pure Ni(OH)₂ spectrum.^{1,2} After calcination, the ratio between the satellite peak and main peak increased, which is in good consistency with the phase transformation of Ni(OH)₂ to NiO characterized by XRD. Typical Ni 2p_{3/2} peak of pure NiO is asymmetrical with an important contribution at ~ 854 eV. For the Ni-sSiO₂&rPMO after calcination, the Ni 2p_{3/2} peak appears symmetrical and shifts to higher binding energy at ~ 856.38 eV, which may be attributed to the strong interaction of Ni²⁺ species with the silica support.^{3,4}

Supplementary Fig. 12 XPS spectra of the obtained Mn-sSiO₂&rPMO (a, b) before and (c, d) after calcination treatment.

Both the Mn 2p spectra show significant multiplet splitting feature⁵. The average oxidation state (AOS) of the Mn centers in the Mn-mJNPs was estimated on the basis of the XPS Mn 3s peak splitting energy (ΔE) using the following correlation⁶:

$$\text{AOS} = 8.95 - 1.13 \times \Delta E \text{ (eV)}.$$

Before the calcination process, the AOS of Mn in Mn-sSiO₂&rPMO is estimated to be 2.28, corresponding to oxidation state of Mn²⁺. After the calcination, the AOS of Mn in Mn-sSiO₂&rPMO was estimated to be 2.74, indicating the transformation of the oxidation state from Mn²⁺ to Mn³⁺, which is in good consistency with XRD results.

Supplementary Fig. 13 (a) XPS spectra of the obtained Co-sSiO₂&rPMO mJNPs before and after calcination treatment. (b) Analysis of XPS spectrum of the Co-sSiO₂&rPMO mJNPs after calcination.

Both Co 2p spectra obtained from the samples before and after calcination show significant multiplet splitting peaks and satellite peaks corresponding to oxidation states of Co. For the Co-mJNP after calcination, the Co 2p_{3/2} peak can be divided into two peaks at 780.58 and 783.60 eV, which corresponding to Co³⁺ and Co²⁺, respectively. So as to the Co 2p_{1/2} peaks, demonstrating the existence of both Co²⁺ and Co³⁺ oxidation state.^{7,8}

Supplementary Fig. 14 XPS spectra of the obtained Cu-mJNPs and Zn-mJNPs. (a) Cu 2p and (b) Cu LMM Auger spectra of Cu-mJNPs before and after calcination. (c) Zn 2p and (d) Zn LMM Auger spectra of Zn-mJNPs before and after calcination.

Both the Cu 2p spectra of Cu-mJNPs before and after calcination show strong satellite peaks at ~ 943 and ~ 962.68 eV, demonstrating the presence of Cu²⁺ in the metal compound.⁹ After the calcination, the multiplet peaks centered at 934.98 and 954.88 eV ascribed to Cu 2p_{3/2} and Cu 2p_{1/2} slightly shift to 935.18 and 955.08 eV accompanied with shape changes, indicating for the change of the binding environments of Cu²⁺, which is consistent with the analysis of the kinetic energy.

As for Zn-mJNP, both the Zn 2p spectra of Zn-mJNPs before and after calcination show distinctive multiplet peaks at ~1022 and ~ 1045 eV ascribed to Zn 2p_{3/2} and Zn 2p_{1/2}, which is consistent with the feature of Zn²⁺.¹⁰ The Auger parameter (calculated with $\alpha' = BE + KE$) of Zn-mJNPs increased by 0.2 eV after calcination, which is in accordance with the transformation between the Zn(OH)₂ and ZnO indicated by XRD results.

Supplementary Fig. 15 XPS spectra of the obtained Y-mJNPs and Gd-mJNPs. (a) Y 3d and (b) O 1s spectra of Y-mJNPs before and after calcination. (c) Gd 3d and (d) O 1s spectra of Gd-mJNPs before and after calcination.

The Y 3d spectra suggest the presence of Y³⁺ in the samples before and after calcination. After calcination, the Y_{3d_{5/2}} and Y_{3d_{3/2}} peaks located at 158.39 and 160.43 eV shifted to lower binding energies of 157.82 and 159.84 eV, indicating the transformation of the metallic compound from Y(OH)₃ to Y₂O₃.¹¹ In addition, the O 1s peak shifted from 531.88 to 530.78 eV, further indicating the transformation of hydroxide compound to metal oxide. Combining with the XRD results, the composition of Y-mJNPs before calcination is considered to be amorphous Y(OH)₃, which transformed to cubic Y₂O₃ after the calcination.

The Gd 4d XPS spectra for both samples show peaks at 142.18 and 147.78 eV corresponding to the trivalent Gd³⁺.^{12,13} The O 1s peak center at 531.18 eV shifted from 530.48 after calcination, indicating the transformation of hydroxide compound to metal oxide.

Supplementary Fig. 18 XPS spectra of the obtained Ca-mJNPs and Mg-mJNPs. (a)

Ca 2p spectrum of Ca-mJNPs. (b) Mg 1s and (c) Si 2p spectra of Mg-mJNPs.

The Ca 2p spectrum obtained from Ca-sSiO₂&rPMO mJNPs shows characteristic peaks corresponding to CaCO₃ with multiplet peaks located at 347.48 (Ca 2p_{3/2}) and 350.98 eV (Ca 2p_{1/2}) and typical satellite loss features at 355.48 and 358.88 eV.¹⁴

As for Mg-mJNPs, the binding energy of Mg 1s in Mg-mJNPs is 1303.88 eV, which is attributed to Mg²⁺ species. The deconvolution of the Si 2p peak of Mg-sSiO₂&rPMO mJNP suggests the presence of two components: one centered at 103.3 eV, attributed to Si 2p from SiO₂, and another one centered at 102.48 eV, which can be assigned to the Si 2p from the MgSiO₃ moieties.¹⁵ It is in good consistency with the XRD analysis of Mg-sSiO₂&rPMO mJNP.

Supplementary Fig. 26 (a) XPS spectrum of Fe 2p and (b) XRD pattern of the obtained Fe-sSiO₂&rPMO mJNPs.

According to the XPS spectrum of Fe 2p, the fitted splitting peaks of Fe 2p_{3/2} and Fe 2p_{1/2} can be divided into two groups: Fe²⁺ (709.8/722.97 eV) and Fe³⁺ (711.13/724.7 eV). The satellite peaks corresponding to Fe shakeup at the high binding energy side of the Fe 2p_{3/2} and Fe 2p_{1/2} were ascribed to Fe³⁺ (719.00/732.37 eV) and Fe²⁺ (713.65/727.21 eV).¹⁶ The XRD patterns further demonstrate that the metal compound is composed by FeO and Fe₂O₃. Based on the analysis of the Fe 2p spectrum via peak fitting, the Fe³⁺/Fe²⁺ ratio in Fe-mJNPs was calculated to be 1.48.

In the revised manuscript, the composites of the obtained M-mJNPs were also analyzed by ICP-OES (Supplementary Table 1): “The metal contents in these representative metal-compound based mJNPs shown in Fig. 2 were measured by inductively coupled plasma-optical emission spectroscopy (ICP-OES) and listed in Supplementary Table 1.”

Supplementary Table 1. The metal contents of representative metal-compound based mJNPs.

Element	Ca	Mg	Mn	Co	Ni	Cu	Zn	Y	Gd
Content (wt %)	5.18	5.41	7.50	6.58	5.22	7.11	10.03	5.97	6.10

Reference:

1. Biesinger, M. C. et al. X-ray photoelectron spectroscopic chemical state quantification of mixed nickel metal, oxide and hydroxide systems. *Surf. Interface Anal.* **2009**, *41*, 324-332.
2. Ang, M. L. et al. Highly Active Ni/xNa/CeO₂ Catalyst for the Water-Gas Shift Reaction: Effect of Sodium on Methane Suppression. *ACS Catal.* **2014**, *4*, 3237-3248.
3. El-Safty, S. A. et al. Nanosized NiO particles wrapped into uniformly mesocaged silica frameworks as effective catalysts of organic amines. *Appl. Catal. A-Gen.* **2008**, *337*, 121-129.
4. Brussino, P. et al. Tuning the properties of NiO supported on silicon-aluminum oxides: Influence of the silica amount in the ODH of ethane. *Catal. Today* **2022**, *394-396*, 133-142.
5. Biesinger, M. C. et al. Resolving surface chemical states in XPS analysis of first row transition metals, oxides and hydroxides: Cr, Mn, Fe, Co and Ni. *Appl. Surf. Sci.* **2011**, *257*, 2717-2730.
6. Mateos, M. et al. Accessing the Two-Electron Charge Storage Capacity of MnO₂ in Mild Aqueous Electrolytes. *Adv. Energy Mater.* **2020**, *10*, 2000332.
7. Bai, X. et al. Hierarchical Co₃O₄@Ni(OH)₂ core-shell nanosheet arrays for isolated all-solid state supercapacitor electrodes with superior electrochemical performance. *Chem. Eng. J.* **2017**, *315*, 35-45.
8. Ning, F. et al. Co₃O₄@layered double hydroxide core/shell hierarchical nanowire arrays for enhanced supercapacitance performance. *Nano Energy* **2014**, *7*, 134-142.
9. Biesinger, M. C. et al. Resolving surface chemical states in XPS analysis of first row transition metals, oxides and hydroxides: Sc, Ti, V, Cu and Zn. *Appl. Surf. Sci.* **2010**, *257*, 887-898.
10. Duchoslav, J. et al. XPS study of zinc hydroxide as a potential corrosion product of zinc: Rapid X-ray induced conversion into zinc oxide. *Corros. Sci.* **2014**, *82*, 356-361.
11. Reddy, I. N. et al. Structural, optical, and XPS studies of doped yttria for superior water splitting under visible light illumination. *J. Electroanal. Chem.* **2019**, *848*, 113335.
12. Nosrati, H. et al. Enhanced In Vivo Radiotherapy of Breast Cancer Using Gadolinium Oxide and Gold Hybrid Nanoparticles. *ACS Appl. Bio Mater.* **2023**, *6*, 784-792.
13. Yin, J. et al. Silica Nanoparticles Decorated with Gadolinium Oxide Nanoparticles for Magnetic Resonance and Optical Imaging of Tumors. *ACS Appl. Nano Mater.* **2021**, *4*, 3767-3779.
14. Moulder, J. F. et al. Handbook of X-ray Photoelectron Spectroscopy, Perkin-Elmer Corp., Eden Prairie, MN, 1992.
15. Brambilla, R. et al. An investigation on structure and texture of silica-magnesia xerogels. *J. Sol-Gel Sci. Techn.* **2009**, *51*, 70-77.
16. Du, W. et al. Fe₃O₄ Mesocrystals with Distinctive Magnetothermal and Nanoenzyme Activity Enabling Self-Reinforcing Synergistic Cancer Therapy. *ACS Appl. Mater. Interfaces* **2020**, *12*, 19285-19294.

Comments 4: Page 10 Figure 4 (b), the caption of "Assembly on PMO&PMO" should

be revised to "sPMO&rPMO".

Response: Thank you for your careful review. We have revised Fig. 4.

Fig. 4 The mechanism of the hydrophilicity mediated interfacial selective growth strategy.

Comments 5: Page 11, lines 264-266. The CA of SiO₂ and PMO was measured on a rod-like structure or sphere structure? Is there any difference? It is also generally accepted that a contact angle <90 degrees is hydrophilic. The CA at 35 degrees should be considered quite hydrophilic as well. Thus it is very questionable to conclude hydrophobicity/hydrophilicity-induced selective synthesis.

Response: Thanks very much for the comment. The contact angle of SiO₂ and PMO in this work was measured on a spherical structure. In the revised manuscript, we have added the contact angle test on a rod-shaped samples. The contact angles measured for rod-like SiO₂ and PMO nanoparticles is ~ 7.6 and ~ 40°, respectively. The results are in good consistency with previous tests measured for spherical silica (6.8°) and PMO (35°), showing that there is no obvious difference between the rod-shaped samples and the spherical samples. In this work, the hydrophilicity is only relative, which means that PMO is more hydrophobic than SiO₂. We further clarify this point in the revised manuscript.

Accordingly, we have added **Supplementary Fig. 24** in the revised **Supporting Information and the corresponding descriptions in the main text of the revised manuscript on Page 12**: “The contact angle of water droplets on SiO₂ surface (~ 6.8° for spherical SiO₂, ~ 7.6° for rod-like SiO₂ nanoparticles) is much smaller than that of on PMO surface (~ 35° for spherical PMO, ~ 40° for rod-like PMO nanoparticles) (Fig. 4c, Supplementary Fig. 24), confirming the different hydrophilicity between SiO₂ and PMO domains.”,

The synthesis methods of the rod-shaped SiO₂ and PMO were also provided in Page 20: “*Synthesis of rod-like silica nanoparticles (rSiO₂):* 100.0 mg of CTAB was dispersed in 50.0 mL of deionized water and 5.0 mL of ethanol by sonication. Then, 1.5 mL of ammonia aqueous solution (28 wt %) was added under stirring. After stirring for 30 min, 600 μL of TEOS was injected and the reaction continued for 9 h. The rSiO₂ nanoparticles were collected by centrifugation, washed with ethanol and water for several times and dried for further use.

Synthesis of rod-like PMO (rPMO): 150.0 mg of CTAB was dispersed in 50.0 mL of deionized water and 3.5 mL of ethanol by sonication. Then, 1.5 mL of ammonia aqueous solution (28 wt %) was added under stirring. After stirring for 30 min, 80 μL of BTEE was injected and the reaction continued for 9 h. The rPMO nanoparticles were collected by centrifugation, washed with ethanol and water for several times and dried for further use.

In order to remove the CTAB surfactant, all the obtained samples went through an extraction process, during which the sample was extracted twice with an alcoholic solution of NH₄NO₃ (6 g/L) at 60 °C and washed three times with ethanol and deionized water.”

Supplementary Fig. 24 (a) Contact angles of water on the rod-like SiO₂ and PMO. TEM images of (b) rod-like silica nanoparticles and (c) rod-like PMO nanoparticles.

The contact angle test was conducted via the sessile drop method. In detail, the sample powder was pressed into a tablet and the contact angle was evaluated from the profile of water droplet on the horizontal solid surface. So, this is a qualitative method to characterize the hydrophilic/hydrophobic properties of the samples. It is difficult to quantify accurately of the surface property at the single-particle level. Thus, the measured value could not accurately quantify such surface property of individual nanoparticles. Although the results for both samples were less than 90°, the difference was very significant. In addition, we also noticed that when adding the sample powder into pure water solution, the silica powder was wetted immediately, while the PMO powder floated on the water surface (**Fig. R1**), which also reflects the significant difference between their hydrophilicities. Also, the PMO materials have been reported to have stronger hydrophobic-hydrophobic interactions with hydrophobic drug molecules than silica in drug delivery, owing to the presence of organic ethyl group in the PMO frameworks.^[1-5] Therefore, we adopt “hydrophilic” to describe silica while use “hydrophobic” to describe PMO for comparative description.

Fig. R1 The photos of the SiO₂ (left) and PMO (right) powders dissolved in the water.

Reference:

1. Asefa T. et al. Periodic meso-porous organosilicas with organic groups inside the channelwalls. *Nature* **1999**, 402, 867-871
2. Munaweera, I. et al. Novel wrinkled periodic mesoporous organosilica nanoparticles for hydrophobic anticancer drug delivery. *J. Porous Mat.* **2014**, 22, 1-10.
3. Croissant, J. G. et al. Syntheses and applications of periodic mesoporous organosilica nanoparticles. *Nanoscale* **2015**, 7, 20318-20334.
4. Guan, B. et al. Highly ordered periodic mesoporous organosilica nanoparticles with controllable pore structures. *Nanoscale* **2012**, 4, 6588-6596.
5. Li, X. et al. Anisotropic growth-induced synthesis of dual-compartment Janus mesoporous silica nanoparticles for bimodal triggered drugs delivery. *J. Am. Chem. Soc.* **2014**, 136, 15086-15092.

Comments 6: Page 15, lines 363-366: The enzymatic kinetic analysis shows that GOx after being functionalized on the Fe-mJNPs retained K_m in the value of 5.3 mM and 8.3 mM, respectively. However, the native GOx purchased from the indicated manufacturer indicated a K_m value of 19 mM. A lower K_m means stronger substrate binding affinity. How can the enhanced substrate binding affinity of the GOx-Fe-mJNPs be explained? Also, the turnover number of K_{cat} (calculated from V_{max}/K_m) for Fe-sSiO₂&rPMO-GOx mJNPs maintained a value of $2 \times 10^{-5} s^{-1}$, while native GOx has a turnover number in the range of 600 to a few thousand per second. Does that mean the enzyme lost its catalytic activity after the EDS-NHS coupling? https://www.aladdin-e.com/zh_cn/g130084.html

Response: Thanks for the valuable question. In fact, the Fe-sSiO₂&rPMO-GOx mJNPs in this work functioned as a nanozyme, which catalyze glucose to generate $\cdot OH$, and there were two catalytic reactions in this system (Fig. R2): I) GOx-catalyzed H₂O₂ generation; II) Fe-sSiO₂-catalyzed $\cdot OH$ generation to induce the TMB oxidation. So, we assume it may not appropriate to compare the catalytic properties of the Janus nanozyme with GOx directly. In the cascade reaction of this work, the catalysis of the

nanozyme is the rate-determining-step, because the catalytic efficiency of the nanozyme is much lower than that of the natural enzyme. So, we would like to propose the comparison with other cascade nanozymes, which can also catalyze glucose to generate $\cdot\text{OH}$. For example, a porous nanozyme of Co-Fc@GOx (Co-ferrocene metal-organic framework combined with GOx) reported by Han group¹, mesoporous nanozyme of GOx-Fe₃O₄@DMSNs (dendritic mesoporous silica nanoparticles (DMSN) integrated with glucose oxidase and ultrasmall Fe₃O₄ nanoparticles) reported by Shi group², etc.

Fig. R2 Schematic illustration of cascade nanocatalysts based on Fe-sSiO₂&rPMO-GOx mJNPs and the catalytic reactions in the catalytic performance evaluation system.

1) For the binding affinity, Han group¹ reported a K_m of 10.6335 mM for Co-Fc@GOx and Shi group² reported a K_m of 10.93 mM for GOx-Fe₃O₄@DMSNs in the similar sequential catalytic reaction by using glucose as the substrate. In our work, both the Fe-sSiO₂-GOx and Fe-sSiO₂&rPMO-GOx show lower K_m value (8.3 mM and 5.3 mM), indicating the stronger substrate binding affinity of the samples.

In our opinion, the stronger substrate binding affinity of the GOx-Fe-mJNPs comes from two aspects i) the exposure of the active catalytic sites and ii) the mesoporous structure of the Janus nanocarrier. Both Fe-sSiO₂&rPMO-GOx and Fe-sSiO₂-GOx possess Fe-based nanosheets structures with high surface area, in which the iron active sites are efficiently exposed to react with the substrates. Moreover, the mesoporous nanoparticles are acknowledged with their high specific surface area and excellent adsorption capacity, which facilitates their binding with the substrates. Thus, the Fe-sSiO₂&rPMO-GOx mJNPs with mesoporous structure can exhibit even higher binding affinity with the substrates.

2) To explore whether GOx would lose its catalytic activity after the EDS-NHS coupling, we assessed the catalytic activity of the GOx before and after the EDS-NHS coupling by using the GOx Activity Assay Kit. And the results show that the GOx retained 62% of its initial activity after the EDS-NHS coupling.

Accordingly, we have added the corresponding descriptions in the main text of the revised manuscript on Page 15: “The GOx activity was evaluated by using GOx Activity Assay Kit, it is found that the GOx retained ~62% of its initial activity after the EDS-NHS coupling.”

The K_{cat} of the native GOx is calculated from V_{max}/[E] ([E] means the concentration of the enzyme). However, the concentration of the nanozyme in our work is hard to quantify because of its complicated structure (including two catalysts, i.e., Fe

and GOx). So, we use V_{max} as a useful parameter to evaluate the catalytic activity of the cascade catalysts. The V_{max} could reach as high as 1.0×10^{-7} M/s in this work, which is higher than the V_{max} (4.22×10^{-8} M/s) reported by Shi group².

Reference:

1. Fang C. et al. Co-Ferrocene MOF/Glucose Oxidase as Cascade Nanozyme for Effective Tumor Therapy. *Adv. Funct. Mater.* **2020**, *30*, 1910085.
2. Huo, M. et al. Tumor-selective Catalytic Nanomedicine by Nanocatalyst Delivery. *Nat. Commun.* **2017**, *8*, 357.

Comments 7: SI: Page 12, Figure S28: certain glucose negative control is missing in order to make full compensation with Figure 5:

Fe-sSiO₂-GOx pH7.4 glucose-

Fe-sSiO₂&rPMO-GOx pH 6.0 glucose-

Fe-sSiO₂&rPMO-GOx pH 7.4 glucose-

Response: Thanks for your suggestion. We have accepted it. In the revised manuscript, the “Fe-sSiO₂&rPMO-GOx pH 6.0 glucose-” group was displayed in Fig. 5g. Four negative control groups “Fe-sSiO₂-GOx pH 7.4 glucose-; Fe-sSiO₂&rPMO-GOx pH 7.4 glucose-; Fe-sSiO₂-GOx pH 6.0 glucose-; Fe-sSiO₂-GOx pH 7.4 glucose+” were added in Supplementary Fig. 39.

Accordingly, we have added Supplementary Fig. 39 in the revised Supporting Information.

Supplementary Fig. 39 CLSM images of ROS probe in the HepG2 cell after different treatments. The cells were co-incubated with the Fe-sSiO₂&rPMO-GOx or Fe-sSiO₂-GOx in the presence and absence of glucose (5 mM) at pH 7.4 and 6.0 for 4 h.

Reviewer #2:

The authors reported a novel pathway for the synthesis of metallic-compound-based mesoporous Janus nanoparticles (mJNPs) via hydrophilicity-mediated interfacial selective assembly. It is noteworthy that, using the sSiO₂&rPMO template (where SiO₂ and PMO were spherical and rod-shaped, respectively), the authors were able to selectively deposit metal onto the hydrophilic SiO₂-subunit to form metallic-compound-based mJNPs (M-mJNPs), in which more than 20 metallic compound-based mJNPs were successfully synthesized. This finding provides a facile synthesis of metallic-compound-based mJNPs and contributes to adding more varieties of Janus materials. Moreover, the as-synthesized Fe-sSiO₂&rPMO-GOx mJNPs were active as a catalyst for enhanced efficient chemodynamic therapy (CDT). Overall, this paper is clearly written and well-organized, and the introduction is reasonable, given the premise of the paper. The data were of high quality, and the analysis was performed very carefully, while the discussion was written in a very compelling way. This work has successfully answered the challenges in synthesizing Janus materials, e.g., achieving a well-defined interface between two different regions, precise control over the distribution of the loaded metal oxides, and creating a broad platform for constructing different functions and structures.

Response: We sincerely thank the reviewer for the positive comments.

Comments 1: However, the present reviewer felt inconvenienced by the term "metallic" in the M-mJNPs, as the term is usually used to describe zero-valent metals and not the oxide or hydroxide forms of metals. Therefore, it is suggested to use a more appropriate term to represent the metal oxides grown on the surface of sSiO₂ of mJNPs.

Response: Thank you for your kind reminder. We borrowed the term “metallic compound” from the literature (*J. Mater. Chem. A* **2021**, 9, 1970-2017)¹ when we were writing this draft. However, the term “metallic” is always represent for zero-valent metals. Now after checking a lot more literatures, we found the term “metal compound” is more commonly used and more appropriate in this case.²⁻⁷ **Accordingly, we revised our manuscript by replacing the word “metallic” with “metal”.**

Reference:

1. Naskar, P. et al. Chemical supercapacitors: a review focusing on metallic compounds and conducting polymers. *J. Mater. Chem. A* **2021**, 9, 1970-2017.
2. Stinn, C. & Allanore, A. Selective sulfidation of metal compounds. *Nature* **2022**, 602, 78-83.
3. Yan, X., Jia, Y. & Yao, X. Defective Structures in Metal Compounds for Energy-Related Electrocatalysis. *Small Struct.* **2020**, 2, 2000067.
4. Zhou, M. et al. First-row transition metal compounds for aqueous metal ion batteries. *J. Energy Chem.* **2021**, 63, 195-216.
5. Sheng, H. Y. et al. Metal-Compound-Based Electrocatalysts for Hydrogen Peroxide Electrosynthesis and the Electro-Fenton Process. *ACS Energy Lett.* **2023**, 8, 196-212.
6. Tran, V. A. et al. Metal-Organic-Framework-Derived Metals and Metal Compounds

as Electrocatalysts for Oxygen Evolution Reaction: A review. *Int. J. Hydrogen Energ.* **2022**, *47*, 19590-19608.

7. Zhang, X. J. & Li, D. Metal-Compound-Induced Vesicles as Efficient Directors for Rapid Synthesis of Hollow Alloy Spheres. *Angew. Chem. Int. Edit.* **2006**, *45*, 5971-5974.

Comments 2: *On the mechanistic study, computational studies on trivalent cations (e.g., Mn^{3+} , Y^{3+} , or Co^{3+}) can be added to confirm the claim on the electrostatic potential distribution of hydrated metal ions on the surface of SiO_2 or PMO.*

Response: Thank you for your useful suggestion, which help us to take into account the difference in the valence states of metal cations. In the revised manuscript, we have conducted computational studies on Y^{3+} to confirm the mechanism. By the way, we apologize for the misleading about the Mn^{3+} and Co^{3+} . In fact, all the transitional metal cations are divalent cations during the assembly process (including Mn^{2+} and Co^{2+} , whose products transformed to be Mn_2O_3 and Co_3O_4 after calcination). And we added the valence state analysis *via* XPS in Supplementary Fig. 12 and Supplementary Fig. 13 in the revised supporting information.

Accordingly, we have added **Supplementary Fig. 25** in the revised **Supporting Information and the corresponding descriptions in the main text of the revised manuscript on Page 13**: “Moreover, the electrostatic potential distribution of $[Y(H_2O)_6]^{3+}$, a representative model of trivalent hydrated cations, on the surface of silica and ethyl-bridged organosilica molecules is calculated (Supplementary Fig. 25a), respectively. Similar to the case of $[Ni(H_2O)_4]^{2+}$, the exposed hydrogen atoms of the $[Y(H_2O)_6]^{3+}$ molecules absorbed on the silica surface have higher peak electrostatic potential energy (~ 11.9 eV) than that of absorbed on the organosilica surface (~ 11.4 eV). And the binding energy of $[Y(OH)(H_2O)_5]^{2+}$ with SiO_2 (-89.00 kcal/mol) is also higher than that of with PMO (-83.82 kcal/mol) (Supplementary Fig. 25b), which contributes to the selective growth of hydrated metal compound on SiO_2 surface.”

Supplementary Fig. 25. (a) The electrostatic potential mappings of $SiO_2-[Y(H_2O)_6]^{3+}$ and $PMO-[Y(H_2O)_6]^{3+}$ composites. (g) The interaction energies of $[Y(OH)(H_2O)_5]^{2+}$ on SiO_2 and PMO.

The valence state analysis *via* XPS were provided in Supplementary Fig. 12 and

Supplementary Fig. 13:

Supplementary Fig. 12 XPS spectra of the obtained Mn-sSiO₂&rPMO (a, b) before and (c, d) after calcination treatment.

Both the Mn 2p spectra show significant multiplet splitting feature⁵. The average oxidation state (AOS) of the Mn centers in the Mn-mJNPs was estimated on the basis of the XPS Mn 3s peak splitting energy (ΔE) using the following correlation⁶:

$$\text{AOS} = 8.95 - 1.13 \times \Delta E \text{ (eV)}.$$

Before the calcination process, the AOS of Mn in Mn-sSiO₂&rPMO is estimated to be 2.28, corresponding to oxidation state of Mn²⁺. After the calcination, the AOS of Mn in Mn-sSiO₂&rPMO was estimated to be 2.74, indicating the transformation of the oxidation state from Mn²⁺ to Mn³⁺, which is in good consistency with XRD results.

Supplementary Fig. 13 (a) XPS spectra of the obtained Co-sSiO₂&rPMO mJNPs before and after calcination treatment. (b) Analysis of XPS spectrum of the Co-

sSiO₂&rPMO mJNPs after calcination.

Both Co 2p spectra obtained from the samples before and after calcination show significant multiplet splitting peaks and satellite peaks corresponding to oxidation states of Co. For the Co-mJNP after calcination, the Co 2p_{3/2} peak can be divided into two peaks at 780.58 and 783.60 eV, which corresponding to Co³⁺ and Co²⁺, respectively. So as to the Co 2p_{1/2} peaks, demonstrating the existence of both Co²⁺ and Co³⁺ oxidation state.^{7,8}

Reference:

5. Biesinger, M. C. et al. Resolving surface chemical states in XPS analysis of first row transition metals, oxides and hydroxides: Cr, Mn, Fe, Co and Ni. *Appl. Surf. Sci.* **2011**, *257*, 2717-2730.
6. Mateos, M. et al. Accessing the Two-Electron Charge Storage Capacity of MnO₂ in Mild Aqueous Electrolytes. *Adv. Energy Mater.* **2020**, *10*, 2000332.
7. Bai, X. et al. Hierarchical Co₃O₄@Ni(OH)₂ core-shell nanosheet arrays for isolated all-solid state supercapacitor electrodes with superior electrochemical performance. *Chem. Eng. J.* **2017**, *315*, 35-45.
8. Ning, F. et al. Co₃O₄@layered double hydroxide core/shell hierarchical nanowire arrays for enhanced supercapacitance performance. *Nano Energy* **2014**, *7*, 134-142.

Comments 3: In the time-dependent absorbance changes of TMB solutions over Fe-sSiO₂&rPMO-GOx mJNPs at different β-D-glucose concentrations (Fig. 5d), an induction period is observed in the initial 20 min for all β-D-glucose concentrations. Thus, it is important to further explain this phenomenon in detail.

Response: Thanks for your comment. We have accepted it. Induction periods, during which the formation of desired products is slow, are often observed in multistep catalytic reactions. In this work, the total catalytic reactions in the test system contains two cascade reaction process: I) GOx-catalyzed H₂O₂ generation; II) Fe-sSiO₂-catalyzed ·OH generation to induce the TMB oxidation (Fig. R3). In this cascade reaction, the catalysis of the nanozyme is the rate-determining-step, because the catalytic efficiency of the nanozyme is much lower than that of the natural enzyme. So, the diffusion of the H₂O₂ generated in the first reaction is an essential step to start the next reaction. The connection of the following steps is considered to be responsible for the generation of induction period: (a) production of H₂O₂ by GOx-catalyzed β-D-glucose reaction; (b) the diffusion of the H₂O₂; (c) the accumulation of H₂O₂ on Fe active sites; (d) generation of ·OH and (e) production of oxTMB by ·OH.

Moreover, we noticed that the induction periods are more obvious in the catalytic reaction on asymmetric Fe-sSiO₂&rPMO-GOx mJNPs than that of on symmetry Fe-sSiO₂-GOx. Comparing the cascade reactions took place on asymmetric and symmetry catalysts, the main difference lies on the diffusion pathway of H₂O₂ from the first reaction site to the position where the second reaction occurs. The longer diffusion pathway of the H₂O₂ on Janus catalyst plays an important role in generating induction period. However, although the induction period is more obvious when catalyzed by Fe-sSiO₂&rPMO-GOx, the catalytic activity is higher ($K_m=5.3$ mM and $V_{max}=1.0 \times 10^{-7}$ M

s⁻¹) due to the depressed side effects.

Fig. R3 Schematic illustration of cascade nanocatalysts based on Fe-sSiO₂&rPMO-GOx mJNPs and the catalytic reactions in the catalytic performance evaluation system.

Accordingly, we have added following descriptions in the main text of the revised manuscript on Page 30: “Induction period, which is often observed in multistep catalytic reactions,⁴⁸⁻⁴⁹ appeared in this cascade catalysis. Longer diffusion pathway of H₂O₂ on Fe-sSiO₂&rPMO-GOx mJNPs plays an important role in producing induction period in this cascade reaction.”

Reference:

48. Benson, S. W. The Induction Period in Chain Reactions. *J. Chem. Phys.* **1952**, *20*, 1605-1612.

49. Guideri, L., De Sarlo, F. & Machetti, F., Conjugate Addition versus Cycloaddition/Condensation of Nitro Compounds in Water: Selectivity, Acid-Base Catalysis, and Induction Period. *Chem. Eur. J.* **2013**, *19*, 665-677.

Reviewer comments, further round review -

Reviewer #1 (Remarks to the Author):

I have no more comments for the revised manuscript and I would highly recommend the manuscript to be published on Nature communications.

Reviewer #2 (Remarks to the Author):

I am satisfied with the revised MS.